# Structural and functional fine mapping of cysteines in mammalian glutaredoxin reveal their differential oxidation susceptibility

Elizabeth M. Corteselli[1], Mona Sharafi[2], Robert Hondal[3], Maximilian MacPherson[1], Sheryl White[4], Ying-Wai Lam[5], Clarissa Gold[5], Allison M. Manuel[1], Albert van der Vliet [1], Severin T. Schneebeli [6], Vikas Anathy[1], Jianing Li [7] ✉ & Yvonne M. W. Janssen-Heininger [1] ✉

Protein-S-glutathionylation is a post-translational modification involving the conjugation of glutathione to protein thiols, which can modulate the activity and structure of key cellular proteins. Glutaredoxins (GLRX) are oxidoreductases that regulate this process by performing deglutathionylation. However, GLRX has five cysteines that are potentially vulnerable to oxidative modification, which is associated with GLRX aggregation and loss of activity. To date, GLRX cysteines that are oxidatively modified and their relative susceptibilities remain unknown. We utilized molecular modeling approaches, activity assays using recombinant GLRX, coupled with site-directed mutagenesis of each cysteine both individually and in combination to address the oxidizibility of GLRX cysteines. These approaches reveal that C8 and C83 are targets for S-glutathionylation and oxidation by hydrogen peroxide in vitro. In silico modeling and experimental validation confirm a prominent role of C8 for dimer formation and aggregation. Lastly, combinatorial mutation of C8, C26, and C83 results in increased activity of GLRX and resistance to oxidative inactivation and aggregation. Results from these integrated computational and experimental studies provide insights into the relative oxidizability of GLRX's cysteines and have implications for the use of GLRX as a therapeutic in settings of dysregulated protein glutathionylation.

Protein S-glutathionylation (PSSG) is a redox-based post-translational modification involving the conjugation of glutathione (GSH) to reactive thiol groups on protein cysteines. PSSG can modulate the activity and function of various classes of redox-sensitive proteins, such as protein tyrosine phosphatases[1], metabolic enzymes[2], and other oxidoreductases[3,4], and therefore has emerged as a regulatory process with the potential to impact diverse (patho)biological pathways. PSSG is controlled by several enzymes responsible for the conjugation to and removal of GSH from protein thiols. The key family of oxidoreductases that catalyze glutathionylation and deglutathionylation reactions are the glutaredoxins (GLRX). In physiological settings wherein the GSH/GSSG redox couple is highly reduced, mammalian

[1]Department of Pathology and Laboratory of Medicine, University of Vermont Larner College of Medicine, Burlington, VT 05405, USA. [2]Department of Chemistry, University of Vermont, Burlington, VT 05405, USA. [3]Department of Biochemistry, University of Vermont, Burlington, VT 05405, USA. [4]Neuroscience Cellular and Molecular Core, University of Vermont Larner College of Medicine, Burlington, VT 05405, USA. [5]Vermont Biomedical Research Network Proteomics Facility, University of Vermont, Burlington, VT 05405, USA. [6]Department of Industrial and Physical Pharmacy and Department of Chemistry, Purdue University, West Lafayette, IN 47907, USA. [7]Department of Medicinal Chemistry and Molecular Pharmacology, Purdue University, West Lafayette, IN 47907, USA. ✉e-mail: jianing-li@pudue.edu; yvonne.janssen@med.uvm.edu

GLRX1 (hereafter referred to as GLRX) acts predominantly as a deglutathionylase[5]. Consequently, the absence of GLRX leads to enhanced PSSG in response to diverse pro-inflammatory mediators, growth factors, or metabolic disease conditions[6–10]. However, GLRXs can also catalyze glutathionylation reactions under highly oxidizing conditions[11]. GLRX has been implicated in the regulation of protein synthesis[12], cellular proliferation[13], apoptosis[14], and inflammatory responses[15], among others. Importantly, GLRX activity has been shown to be attenuated in various disease settings and in response to cigarette smoke[7,8,16]. However, the precise mechanism(s) that govern the diminished GLRX activity in these situations remains unclear.

As members of the thioredoxin protein superfamily, GLRXs contain a classic thioredoxin-type fold consisting of three α-helices flanked by a four-stranded β-sheet. Mammalian GLRX contains two cysteines in a "C-X-X-C" active site motif, allowing for the reduction of protein disulfides and deglutathionylation. GLRX performs deglutathionylation in a "ping-pong" mechanism, the first step of which is the nucleophilic displacement of GSH from a glutathionylated protein by the N-terminal active site cysteine[17,18]. The rapid rate of this half reaction is partially attributed to the low $pK_a$ (approximately 3.5) of the N-terminal thiol[11]. The resulting glutathionylated GLRX is then reduced by a second molecule of GSH. The glutathione system, specifically GSH, glutathione reductase (GR), and NADPH, provide reducing equivalents to fuel these reactions[19] (Fig. 1A).

GLRX contains 5 cysteines (Fig. 1B, C), and several studies have examined the functional importance of some of these cysteines for GLRX's function. It is well established that GLRX loses all activity upon mutation of the N-terminal active site cysteine[8,20,21]. In contrast, mutation of the C-terminal active site cysteine increases the activity of GLRX, presumably by preventing disulfide bond formation between the two active site cysteines[20,21], and renders GLRX a monothiol enzyme only capable of catalysis of (de)glutathionylation[22]. The three cysteines outside the active site (C8, C79, C83) are also vulnerable to oxidation, and both intra- and inter-molecular disulfide bond formation have been reported[8,23–25]. However, the aforementioned studies did not elucidate the relative propensities of GLRX's cysteines to react with $H_2O_2$ or GSSG, nor did they reveal how oxidation of each cysteine, either alone or in combination, contributes to inactivation of GLRX. In addition, the majority of prior studies examining GLRX activity used assays that incorporated non-physiological substrates, an excess of NADPH, GR, and GSH, and measured the oxidation of NADPH as a proxy for GLRX activity.

Spectroscopic techniques have provided structural information about GLRX[26–28]. Molecular modeling studies represent powerful approaches to provide detailed mechanistic insight into GLRX's structure and reactivity with oxidants or GSH. Molecular dynamics (MD) simulations have yielded critical insights into GSH binding to GLRX2, GLRX5, and ScGLRX7[29–31]. In the present study, we utilized quantum and classical modeling to predict the reactivity of each GLRX cysteine with GSSG and $H_2O_2$, and their implications for aggregation. The relative propensities of GLRX's cysteines to react with GSSG were confirmed experimentally using mass spectrometry. Furthermore, the susceptibility of WT or Cys-to-Ser mutants of GLRX to $H_2O_2$- or GSSG-mediated inactivation was evaluated by measuring the reaction of GLRX with eosin-labeled GSSG or eosin-conjugated glutathionylated BSA, to model physiological substrates. Results from these multi-scale approaches reveal that cysteines 8 and 83 are the main targets of glutathionylation and oxidation that lead to loss of activity, while cysteine 79 is less vulnerable to oxidation. Our collective findings provide a greater understanding into the mechanisms that govern oxidative inactivation of GLRX and may have implications for disease conditions in which PSSG and/or GLRX status are dysregulated.

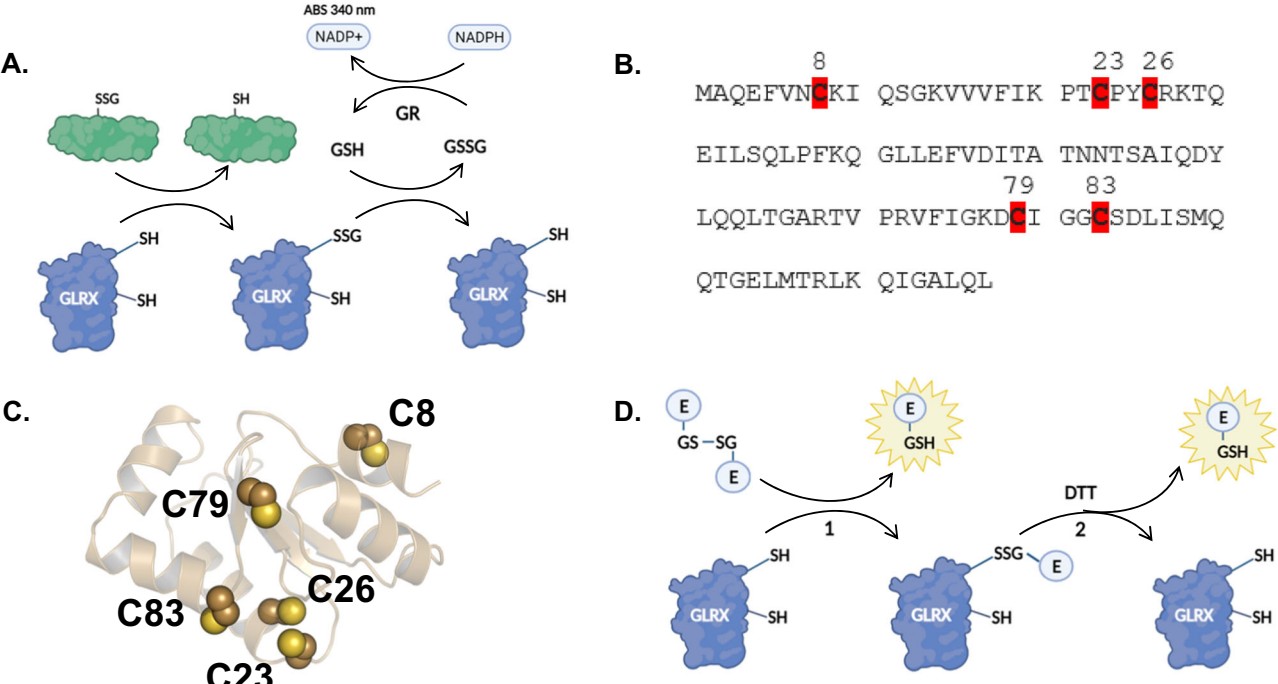

**Fig. 1 | Overview of GLRX catalysis and activity assay used in this study.**
**A** Monothiol mechanism of GLRX, in which GLRX becomes itself glutathionylated during deglutathionylation of a protein target (green). The resulting oxidized GLRX is then reduced by GSH, which is supplied through the GR/NADPH/NADP+ system. **B** Amino acid sequence of murine GLRX1 with mutated amino acids highlighted in red. **C** Ribbon structure of murine GLRX with cysteines labeled. **D** Schematic of fluorescent di-eosin(diE)-GSSG assay used in this study. In reaction 1, diE-GSSG is reduced by GLRX, which is itself glutathionylated and the fluorescent Eosin-GSH is released. In the presence of GSH or DTT, reaction 2 can occur, in which GLRX is reduced and another molecule of Eosin-GSH is released. **A, D** were created with biorender.com.

## Results

### Molecular analysis reveals structural stability of GRLX and pKa of individual cysteines

Given the previous observations that GLRX can be inactivated via an oxidative mechanism[8,23], herein we sought to investigate the relative vulnerability of each of the individual cysteines towards GSSG, a model glutathionylating agent, or $H_2O_2$ and their contribution to oxidative inactivation. We utilized site-directed mutagenesis of mouse GLRX in which one or multiple cysteines were mutated to serine (Supplementary Figs. 1 and 2). To ensure that that each mutation did not significantly alter the tertiary structure of GLRX, we carried out 400-ns-long MD simulations. Analysis of root-mean-square deviation (RMSD) of GLRX over the course of MD simulations before and after each mutation confirmed that the overall fold of GLRX was not altered by these mutations (Supplementary Fig. 5). In addition, no significant conformational change of GLRX was observed upon formation/breakage of C23-C26 disulfide bond within 400 ns. Therefore, despite being a small protein, the three-dimensional structure of GLRX is stable, with tolerance to various cysteine substitutions and redox states.

We next examined the $pK_a$ values of the five cysteines: C8, C23, C26, C79, and C83, as cysteine thiol deprotonation is a critical step for subsequent glutathionylation or oxidation by $H_2O_2$[32]. In our calculations (Supplementary Fig. 6) using PROPKA, C23 was determined to be the most acidic cysteine ($pK_a = 5.4$) followed by C8 ($pK_a = 7.4$), both of which are more acidic than the thiol group in GSH ($pK_a = 8.7$); the other cysteines are more basic (C79 and C83, $pK_a = -11$; C26, $pK_a = 16.7$). Although the error bar of PROPKA on cysteine $pK_a$ ($\pm 3.9$)[33] indicates the qualitative nature of our results, our finding of C23 as the most acidic among all five cysteine is in agreement with previous reports that it plays a key role in GLRX's enzymatic activity[34].

### Mutation of cysteine 26 alone or in combination with mutation of other cysteines increases GLRX deglutathionylation activity

We next investigated the deglutathionylating activity of WT GLRX or GLRX Cys-Ser mutants using diE-GSSG (Fig. 1D, reaction 1) or E-GS-BSA, the latter modeling an S-glutathionylated protein[35]. These assays were conducted in the absence of GSH, GR, or NADPH, but instead included 5 μM DTT as a reductant to reduce GLRX following its reaction with diE-GSSG (Fig. 1D, reaction 2). We chose to use low concentration of DTT rather than GSH/GR/NADPH to minimize interference by these GSH system components with oxidized versions of GLRX. This concentration of DTT did not reduce the diE-GSSG substrate, consistent with a prior report[36]. Kinetic analyses with varied concentrations of WT GLRX and diE-GSSG substrate demonstrated linear GS-eosin formation, and calculated a Km of 28.9 nM and Vmax of 7.03 nM/min (Fig. 2A, B). We next comparatively assessed the rate of reaction of WT or GLRX mutants with diE-GSSG or E-GS-BSA. In agreement with previous studies, mutation of C23 resulted in no detectable catalytic activity (Fig. 2C)[20,21]. However, the C26S mutant or combinatorial mutants containing the C26S mutation displayed significantly faster deglutathionylating activity than WT GLRX protein using E-GS-BSA as the substrate (Fig. 2D, F) with similar trends observed using diE-GSSG (Fig. 2C, E).

### Cysteines 8 and 83 are important for GSSG-induced inactivation of GLRX

We and others have demonstrated that GLRX can be glutathionylated in association with diminished enzyme activity[8,23]. To further improve our understanding of which cysteines in GLRX react with GSSG, we carried out Induced Fit Docking (IFD) of GSSG to the GLRX model at the proximity of each cysteine. Given the lack of deep binding pockets on GLRX, GSSG will likely form extensive contacts with the protein domain containing C23 (Fig. 3B). In addition to the docking score and the RMSD of GSSG, we also considered the S-S distance between GSSG

and each cysteine residue in GLRX, as a short distance is required for the consequent glutathionylation process. Among all the poses from IFD, docked GSSG on C23 represents many polar contacts with several nearby residues on GLRX protein including hydrogen bonding with C83, N55, Y25, G82, P24, T22, and K20 which stabilizes the complex significantly. Therefore, the docked GSSG on C23 provides highest docking score (−7.1 Kcal/mol, Fig. 3F) along with the lowest S-S distance (3.7 and 4.2 Å), and hence reflects the best candidate for glutathionylation, among other cysteines. C23 was followed by C8 (Fig. 3A) and C83 (Fig. 3E) establishing less contacts relative to C23. C8 is also readily accessible for the reaction with bulky GSSG. Finally, the more buried cysteines, C79 (Fig. 3D) and especially C26 represent the highest S-S distance (Fig. 3C), making the disulfide bond formation between GSSG and C79 or C26 less likely to occur. Our docking results indicate a potential hierarchy in relative susceptibilities of GLRX's cysteines towards GSSG-induced glutathionylation, which is C23 > C8, C83 ≫ C79, C26 (Fig. 3F).

We next utilized mass spectrometry to measure glutathionylation of WT GLRX following incubation with 1 or 50 mM GSSG. Glutathionylation of C23 (peptide VVVFIKPTCPYCR), and C8 (peptide MAQEFYNCK) was readily detectable following incubation with 1 mM GSSG, while glutathionylation of C79 and C83 (peptide DCIGGCSDLISMQQTGELMTR) was more prominently detected in response to 50 mM GSSG. Contrasting our IFD prediction, C26 (peptide VVVFIKPTCPYCR) also was readily glutathionylated (Fig. 4A, B and Supplementary Fig. 3). Although comparisons of abundance between different tryptic peptides are not feasible due to differences in ionization efficiencies of each peptide, these results confirm that glutathionylation of each cysteine in WT GLRX occurs following incubation with GSSG, and suggest that C23, C26, and C8 are more readily glutathionylated at lower concentrations of GSSG.

Finally, we addressed the contributions of GLRX' cysteines in glutathionylation-induced inactivation. In agreement with previous reports, WT GLRX lost approximately half of its activity following incubation with GSSG[8], which was restored by subsequent incubation with DTT (Supplementary Fig. 7). Mutation of C8, C26, or C83 conferred protection against the PSSG-mediated decrease in activity towards diE-GSSG, yielding activities that were not statistically different from the respective controls. In contrast, mutation of C79 did not protect against GSSG-mediated inactivation (Fig. 4C). Combinatorial mutants, including C26/C79S, were refractory to GSSG-mediated diminished activity. These results demonstrate that C8, C26, and C83 are the main cysteines involved in GSSG-mediated loss of GLRX activity.

### Cysteines 8 and 83 contribute to oxidative inactivation of GLRX by $H_2O_2$

The formation of sulfenic acid residues is considered the gateway cysteine oxidation induced by $H_2O_2$ that can give rise to other oxidative modifications, including glutathionylation. We next applied quantum mechanics (QM) calculations to obtain greater insight into the reaction mechanism of GLRX with $H_2O_2$. Oxidation of reactive cysteines represents a multi-step process involving a deprotonated thiol and the reaction of active thiolate with $H_2O_2$[37–41] (Supplementary methods). A hybrid quantum mechanics/molecular mechanics (QM/MM) method was established to model the reaction energies of each cysteine with $H_2O_2$ (Supplementary methods) According to this method, the activation energy ($E_a$) for oxidization of deprotonated GLRX cysteine residues follows the following trend: $E_a(C8) \simeq E_a(C83) < E_a(C26) \simeq E_a(C23) < E_a(C79)$ (Fig. 5F). Notably, the $E_a$ for oxidation of the methanethiolate model compound (21 kcal/mol) is higher than the activation energies for C8 ($E_a(C8) = 8.8$ kcal/mol) and C83 ($E_a(C83) = 9.9$ kcal/mol) in GLRX, which indicates a stabilizing effect of the protein environment on the transition state of reaction. This finding is in agreement with earlier studies[42], which demonstrated that the protein environment

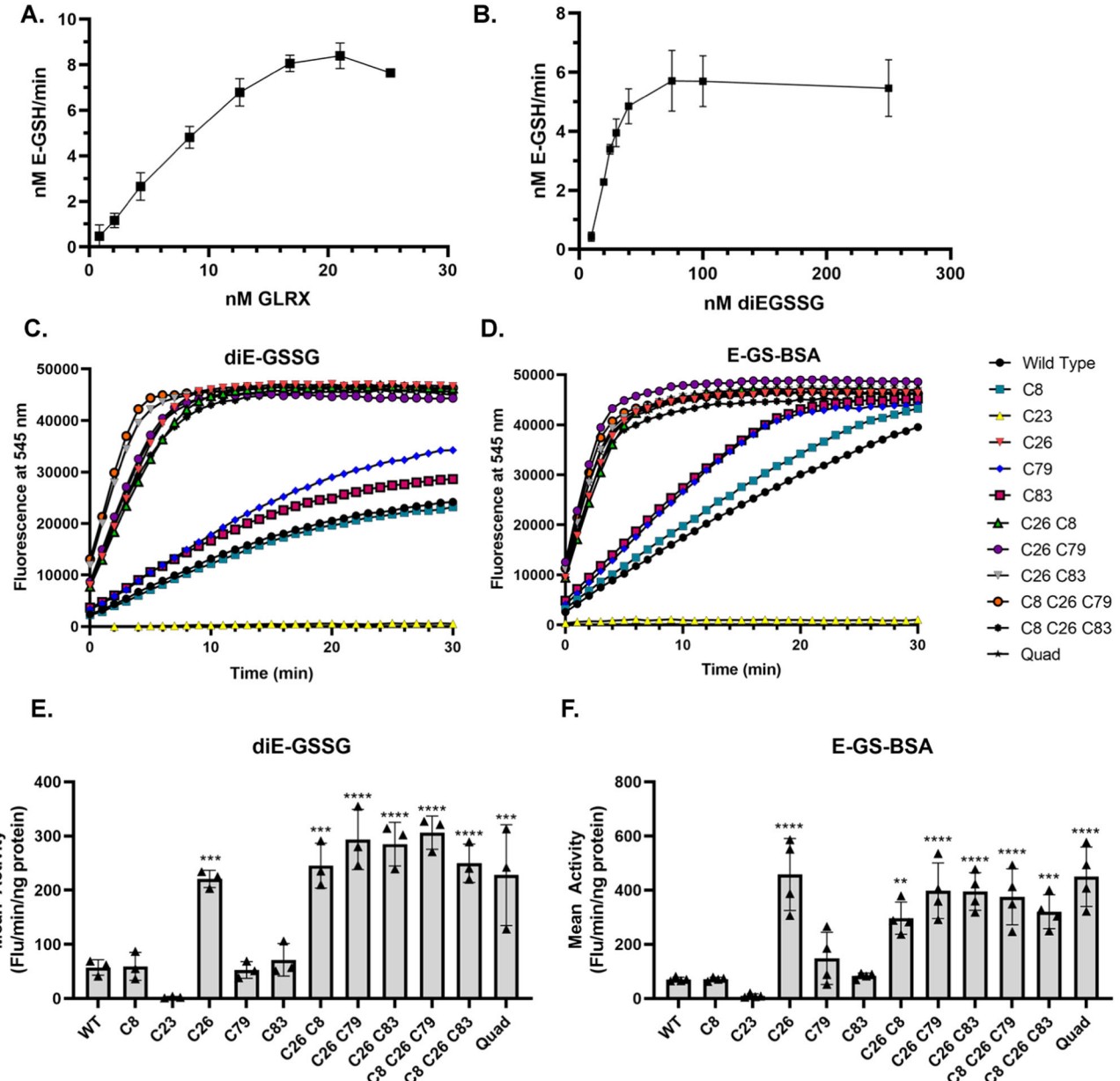

**Fig. 2 | Mutation of C26, alone or in combination, increases activity of GLRX.**
**A, B.** Kinetic analysis of WT GRLX against diE-GSSG substrate, varying either the enzyme (**A**) or substrate (**B**) concentration. Km and Kcat were calculated using Kaleidagraph (5.01). Data are presented as mean ± SD, $n = 3$. **C–F** Comparative assessment of WT and Cys-Ser GLRX mutants utilizing diE-GSSG (**C, D**) or diE-GS-

BSA (**E, F**) as the substrate using 20 nM GLRX. Mean activity was calculated as the slope of the linear portion of each curve normalized to GLRX protein (**C, D** 25 ng). Mean ± SD, $n = 3$. Comparisons performed using one-way ANOVA with multiple comparisons and Dunnett correction, $*p \leq 0.05$, $**p \leq 0.01$, $***p \leq 0.001$, $****p \leq 0.0001$.

indeed changes the susceptibility of cysteines to oxidation. C8 and C83 display the highest tendency for oxidation ($E_a$(C8) = 8.8 kcal/mol, $E_a$(C83) = 9.9 kcal/mol) in their reaction with $H_2O_2$. On the other hand, C79 shows the lowest tendency for oxidation ($E_a$(C79) = 30.6 kcal/mol), while C26 and C23 show intermediate tendencies for oxidation ($E_a$(C23) = 19.3 kcal/mol, $E_a$(C26) = 17.6 kcal/mol) (Fig. 5F). Transition state conformational analysis of C8 (Fig. 5A) indicates that $H_2O_2$ is able to form hydrogen bonds with the carboxylates of I10 and N7 as well as with the thiolate of C8 (which can further stabilize the negative charge on sulfur atom of C8). Lastly, the oxygen atoms on $H_2O_2$ also from hydrogen bonds with the $CONH_2$ group of Q11. These hydrogen bond interactions, in addition to significant accessibility of C8 to solvent, play a key role in stabilizing the TS and can reduce the $E_a$ by ~12 kcal/mol, compared to the $[CH_3S\cdots HOOH]^-$ model transition state, which lacks these interactions.

Like C8, the transition state obtained for C83 also explains the observed reduction of the $E_a$ relative to the $[CH_3S\cdots HOOH]^-$ model transition state. Specifically, during C83 oxidation, P71 and Y25 as well as the $CONH_2$ of C83 form stabilizing hydrogen bonds (Fig. 5E) with the thiolate and the $H_2O_2$ substrate in the transition state. Furthermore, the hydrophobic environment around C79, mainly F74, likely increases the energy cost of the reaction since the hydrophobic F74 is not well suited to stabilize the negatively charged transition state of the oxidation reaction (Fig. 5D). Finally, C23 and C26 (Fig. 5B, C) demonstrate comparable oxidative behavior, which is not surprising as they reside in similar environments inside of the protein's binding pocket. However, C26 is more buried, with enhanced capacity to establish stabilizing hydrogen bonding interactions with the surrounding residues (K20 and C23 in particular). Overall, the results obtained from the QM/MM calculations point to a relative propensity of $H_2O_2$-induced

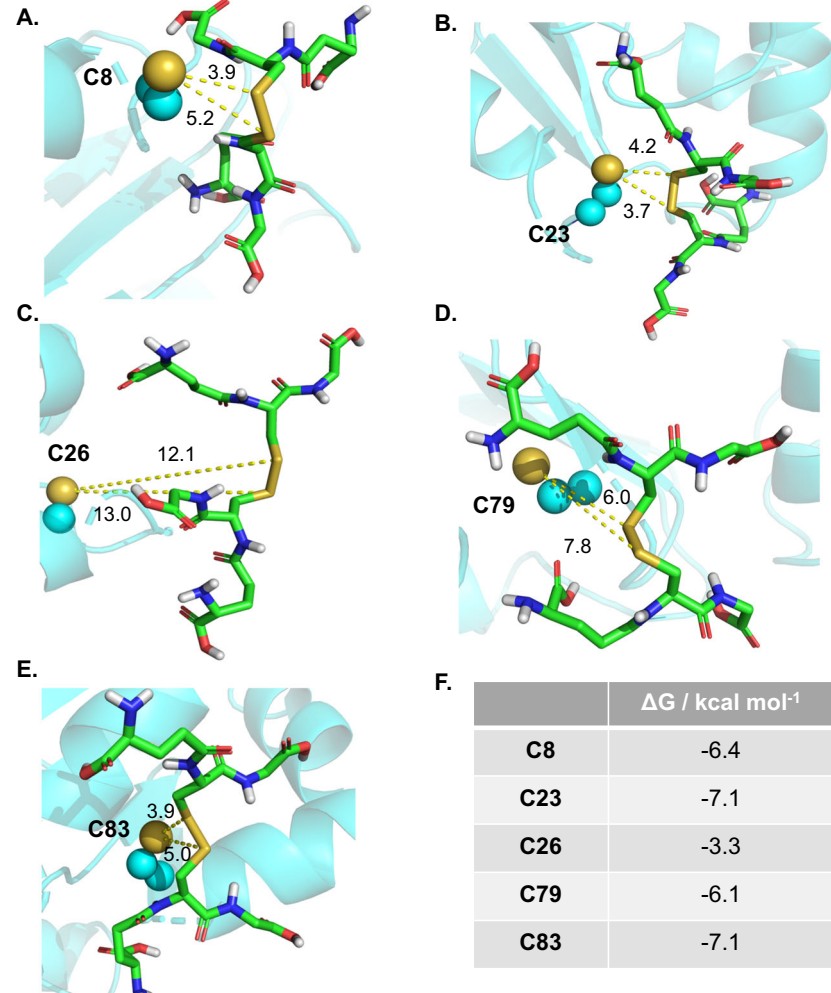

**Fig. 3 | Molecular docking reveals hierarchy of reactivity of GLRX cysteines with GSSG.** The docking pose corresponds to the lowest S-S distance of GSSG docked to the proximity of **A** C8. **B** C23. **C** C26. **D** C79 and **E** C83. The sulfur groups on cysteines are shown as yellow spheres and the GSSG ligand is shown in sticks. The yellow dashed lines show the S-S distances labeled with numbers in Å. **F** Summary of docking scores for each panel. The reported scores are the ones of the poses representing the lowest distance between S-S groups. **All numbers in Panels A–E are distances in angstrom.

oxidation of GLRX preferentially targeting C8 and C83, with resultant formation of sulfenic acid (SOH) and potential sulfenylamide intermediates, as well as more oxidized sulfinic (−SOO$_2$H) and sulfonic acids (−SOO$_3$H) which can form under severe oxidative stress.

Finally, we addressed the ability of H$_2$O$_2$ to diminish GLRX activity and the contributions of various cysteines in this process. For this purpose, WT or Cys-Ser mutants of GLRX were incubated with 100 μM H$_2$O$_2$, and excess H$_2$O$_2$ was removed using catalase. As shown in Fig. 5G, the activity of WT GLRX was reduced by ~30% following oxidation by H$_2$O$_2$. In contrast, mutation of C8 or C83 conferred protection against H$_2$O$_2$-induced inactivation. Notably, several of the GLRX compound mutants containing C8S or C83S, such as the C8S/C26S/C83S and quadruple mutants, also showed significantly diminished activity following incubation with H$_2$O$_2$, potentially reflecting overoxidation of the remaining cysteine, C23, contributing to loss of activity, or other amino acids entirely.

## Cysteine 8 plays an important role in the formation of GLRX dimers during oxidation

The ability of GLRX monomers to form aggregates has been previously demonstrated and is speculated to occur through formation of disulfide bonds between cysteines of separate monomers[23]. To further elucidate the impact of each individual cysteine on GLRX dimerization,

we simulated two GLRX dimer models. The first model was constructed by aligning the GLRX model (PDB ID: 4RQR) to a zebrafish GLRX2 dimeric structure (PDB ID: 3UIW), showing the C8-C8 separation as short as 1.0 Å (measured between the sidechain S atoms). We simulated this GLRX dimer model (Supplementary Fig. 8) for 200 ns, and observed a stable, symmetric interface with the S-S distance adjusted to 8.9 Å (Supplementary Movie 1). In parallel, we also simulated three free GLRX monomers in solution (Fig. 6A). Within 200 ns, we observed the formation of the C8-C8 interface (the S-S distance at 9.5 Å) which approximates the one from the alignment model, in addition to an asymmetric C8-C23 interface, with the S-S distance at 6.5 Å (Fig. 6A, Supplementary Movie 2). Compared to other cysteines in GLRX, C8 is largely exposed to the protein surface. We also found the symmetric and asymmetric interfaces involving C8 to have good shape complementarity. Notably, GLRX like many soluble proteins, has a charged protein surface and almost no hydrophobic patch. Therefore, these simulations provide direct evidence to support a significant involvement of C8 in the GLRX monomer-monomer interfaces, and the simulated S−S distance between C8 residues suggests a likelihood of crosslinking as well as shape complementarity to drive the dimer formation.

We next used gel electrophoresis of WT and GLRX Cys-Ser mutants to address the involvement various Cys residues in GLRX

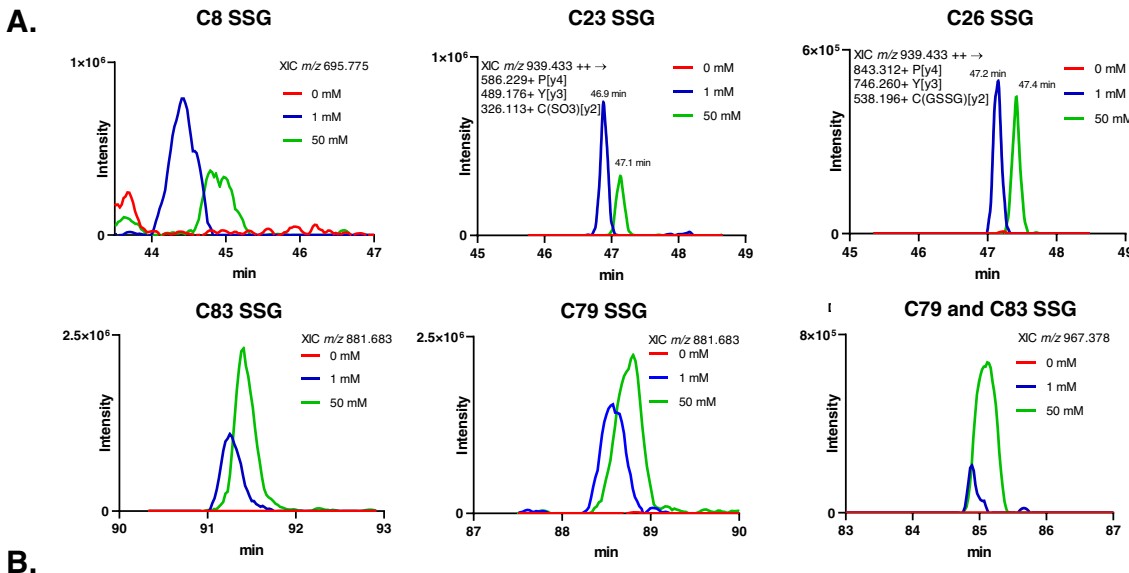

**B.**

| Peptide | Modification | % Area under the curve ± SD | | |
|---|---|---|---|---|
| | | 0 mM | 1 mM | 50 mM |
| MAQEFVNCK | C8 glutathionylated | 1.1 ± 0.7 | 56.9 ± 7.7 | 42.0 ± 8.4 |
| VVVFIKPTCPYCR | C23 glutathionylated | 0.1 ± 0.1 | 65.1 ± 5.2 | 34.8 ± 5.0 |
| VVVFIKPTCPYCR | C26 glutathionylated | 0.1 ± 0.1 | 56.1 ± 9.2 | 43.7 ± 9.1 |
| DCIGGCSDLISMQQTGELMTR | C79 glutathionylated | 0.1 ± 0.1 | 39.3 ± 0.3 | 60.6 ± 0.3 |
| DCIGGCSDLISMQQTGELMTR | C83 glutathionylated | 0.1 ± 0.1 | 38.9 ± 1.9 | 61.0 ± 2.0 |

**C.**

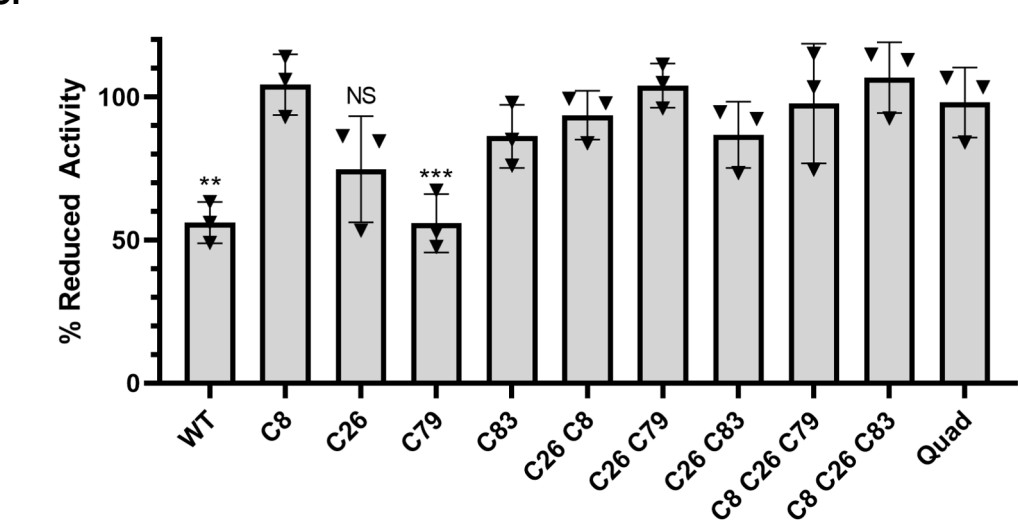

**Fig. 4 | C8 and C83 are targets for glutathionylation and resulting inactivation of GLRX. A** Recombinant GLRX was incubated with 0, 1, or 50 mM GSSG. After alkylating the unmodified cysteines and trypsin digestion, GLRX peptides were detected and quantified by mass spectrometry. Extracted ion chromatographic traces are shown for targeted glutathionylated peptides. The identity of each peptide was confirmed by the MS/MS scans collected during the peak elution (Supplementary Fig. 3). **B** Percentage of area under the curve for each glutathionylated peptide across 0, 1, and 50 mM GSSG concentrations. Data is presented as mean ± SD, $n = 2$. **C** Percentage of reduced activity remaining following incubation with 50 mM GSSG for each mutant. GSSG was removed prior to measurement of activity with di-eosin-glutathione disulfide (diE-GSSG). Mean ± SD, $n = 3$. Statistics performed using multiple unpaired one-way t-tests with a Holm-Sidak correction between % remaining activity and reduced activity (100%) for each mutant, $*p \leq 0.05$, $**p \leq 0.01$, $***p \leq 0.001$, $****p \leq 0.0001$. NS: not significant.

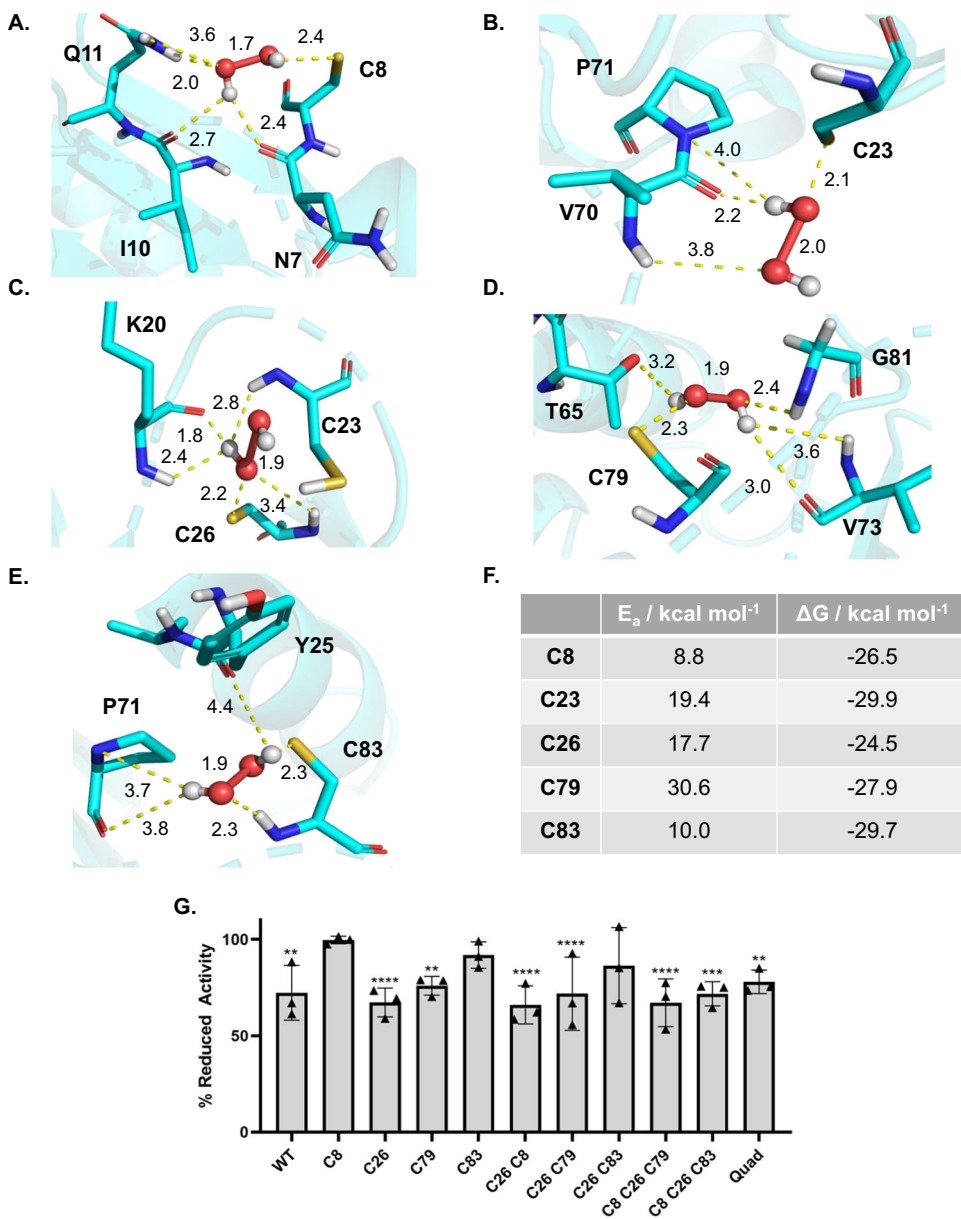

**Fig. 5 | Quantum Mechanics/Molecular Mechanics (QM/MM) modeling of GLRX cysteines reveal that C8 and C83 are most reactive with $H_2O_2$.** Transition state of GLRX/$H_2O_2$ system for all the cysteines and representation of the stabilizing interactions in protein environment. **A** C8. **B** C23. **C** C26. **D** C79, and **E** C83. The labels of dash lines show distances in Å. **F** $E_a$ and $\Delta G$ for each cysteine residue are indicated. **G** Mutation of GLRX cysteine 8 and 83 protects against oxidative inactivation from $H_2O_2$. GLRX mutants were incubated with 100 μM $H_2O_2$ for 10 min and $H_2O_2$ was removed by treatment with catalase prior to measurement of activity with diE-GSSG. Results are expressed as percentage of reduced activity compared to untreated controls for each mutant. Mean ± SD, $n = 3$. Statistics performed using multiple unpaired one-way t-tests with a Holm-Sidak correction between % remaining activity and reduced activity (100%) for each mutant, *$p \le 0.05$, **$p \le 0.01$, ***$p \le 0.001$, ****$p \le 0.0001$., *$p \le 0.05$, **$p \le 0.01$, ***$p \le 0.001$, ****$p \le 0.0001$.

dimer formation. When mutants were separated under reducing conditions, a single band at the predicted molecular weight of GLRX (12 kDa) was detected (Fig. 6B). However, under aerobic non-reducing conditions a band around 22 kDa, the approximate molecular weight of a dimer, was observed for some of the mutants (Fig. 6C). Mutation of C8, as a single mutation or in combination, eliminated this formation of this species, pointing to the importance of C8 in dimer formation. Taken together, these data from molecular simulations and recombinant proteins support that C8 is primarily responsible for GLRX dimer formation in an aerobic environment.

To elucidate whether glutathionylation affected dimerization of GLRX, we conducted non-reducing SDS-PAGE and Western blotting for GLRX. WT GLRX under control aerobic conditions showed some dimer formation (Fig. 7A), likely due the oxidation during sample preparation

in room air, and this was not affected by treatment with GSSG. In contrast, C8S mutant GLRX was refractory to dimerization, consistent with earlier observations (Fig. 6C[23]). C26S, C79S, C83S, C26S/C79S, and C26S/C83S GLRX showed increased dimer formation compared to WT GLRX in the presence or absence of GSSG, and a small shift in MW was observed in response to GSSG, potentially reflecting the addition of GSH. GSSG incubation of C26S and C26S/C79S GLRX also yielded apparent trimers indicative of aggregation (Fig. 7A). Combinatorial mutation of C8/26/79, C8/26/83, or C8/C26/79/83 diminished GSGG-induced di/trimerization. Last, we examined the extent to which $H_2O_2$-mediated oxidation promotes aggregation of WT GLRX or GLRX mutants using western blotting. Results in Fig. 7B demonstrate that incubation with $H_2O_2$ increased dimer and higher molecular weight oligomer formation in WT GLRX as well as single GLRX C26S, C79S,

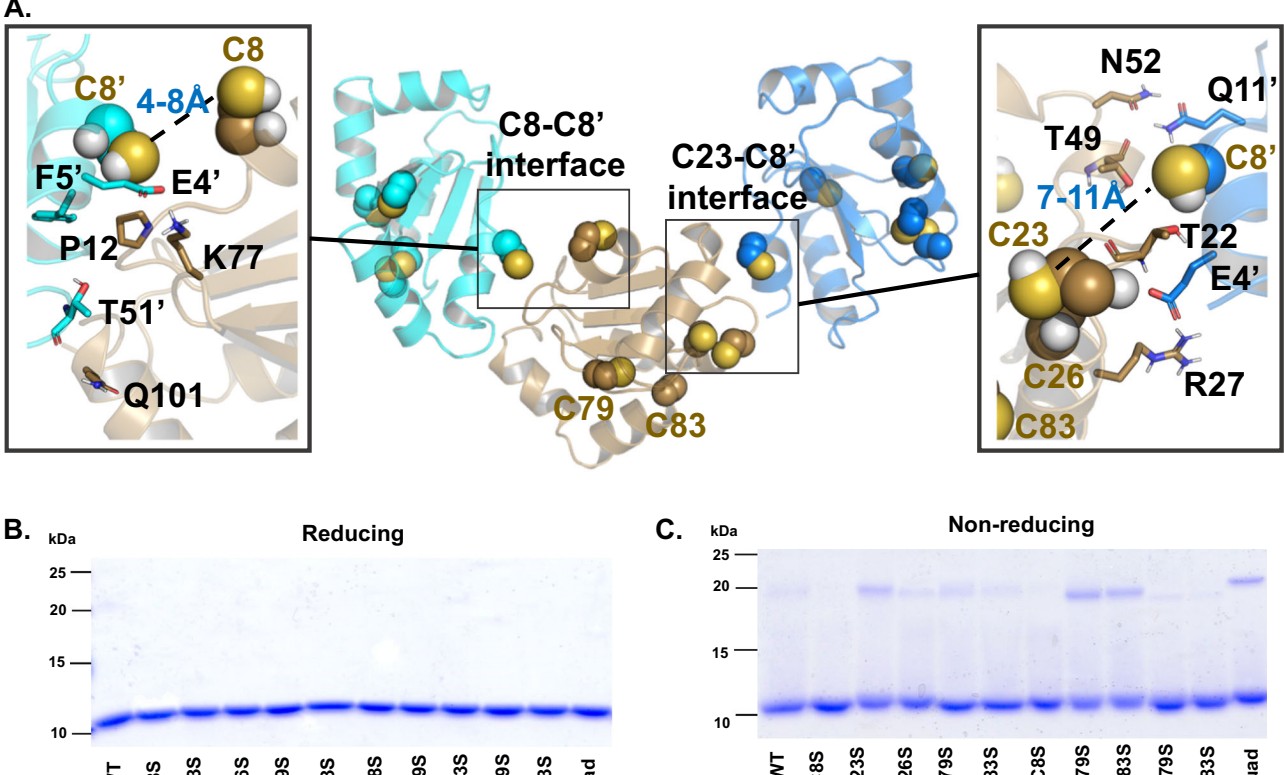

**Fig. 6 | Mutation of C8 reduces dimer formation under non-reducing conditions. A** 200-ns MD simulation of three GLRX monomers in solution. Each monomer is shown in different color and cysteines involved in asymmetric and symmetric interfaces are shown as yellow spheres. See supplementary Movies 1 and 2 for the corresponding trajectories. Reducing (**B**) and non-reducing (**C**) gels of GLRX mutants stained with Coomassie Blue. 10 µg of protein was loaded per lane. Gel is representative of $n = 3$ independent experiments.

and C83S mutants. However, the presence of higher molecular weight oligomers was diminished in C8S mutant, and mutants containing C8S mutation, particularly the quadruple mutant that only contains catalytic C23 (Fig. 7B), similar to the prior results with GSSG (Fig. 7A). These findings collectively support a role of C8 in oxidant-induced dimerization and aggregation of GLRX, while C26, C79, and C83 play complex roles in governing dimerization and/ aggregation of GLRX under oxidizing conditions.

## Discussion

The conjugation and removal of GSH from protein thiols is a highly regulated and critical biological process. S-glutathionylation affects all major classes of proteins, controls the activity of both thioredoxin and peroxiredoxin redox systems (reviewed in ref. 9) and provides protection from irreversible overoxidation by allowing restitution of the original thiol group via the glutaredoxin system[43]. GLRX regulates the balance of glutathionylated proteins by performing deglutathionylation with greater efficiency than any other cytosolic oxidoreductase, including thioredoxins, in physiological settings[44]. Although GLRXs themselves are vulnerable to oxidative inactivation, the relative importance of GLRX' five cysteines in the regulation of GLRX's activity and susceptibility to oxidation remained elusive. Utilizing novel molecular calculations and experimental validation of murine GLRX using an assay that was modified for this purpose, in the present study we demonstrate the contribution of C8 and C83 to oxidative inactivation and of C8 to oxidant-induced aggregation. Through the design of combinatorial mutants involving C8, C26, and C83, we show that

mutation of these cysteines to serines yields a GLRX molecule with increased deglutathionylating activity and resistance to oxidative inactivation. Lastly, GLRX containing mutations of all four non-catalytic cysteines retains deglutathionylating activity and is largely refractory to oxidant-induced protein aggregation.

While it had been previously demonstrated that GLRX can be inactivated by oxidants[23], it remained unclear which cysteines are the most prone to $H_2O_2$- or GSSG-induced oxidation and what consequence this may have specifically for its deglutathionylation activity. In the present study, we applied for the first time a combined strategy consisting of molecular simulations and hybrid QM/MM calculations to gain molecular understanding of the structure-function relationships of all GLRX cysteine residues. In particular, we placed emphasis on delineating the impact of $H_2O_2$-linked oxidation of GLRX by calculating the reaction thermodynamical activation parameters for all the cysteine residues present. Using both induced fit docking as well as QM/MM calculations we determined that C8 and C83 most easily react with $H_2O_2$ and GSSG, findings in agreement with experimental results showing that mutation of these two cysteines confers resistance to loss of activity from GSSG or $H_2O_2$. Additionally, QM/MM calculations determined that C23 has only an intermediate tendency for oxidation by $H_2O_2$. This is in agreement with previous reports that GLRX is a relatively poor peroxidase in comparison to glutathione peroxidase[45]. Analysis of GLRX incubated with GSSG by mass spectrometry revealed that all cysteines were found to be glutathionylated and largely supported the relative reaction propensities with GSSG that were determined

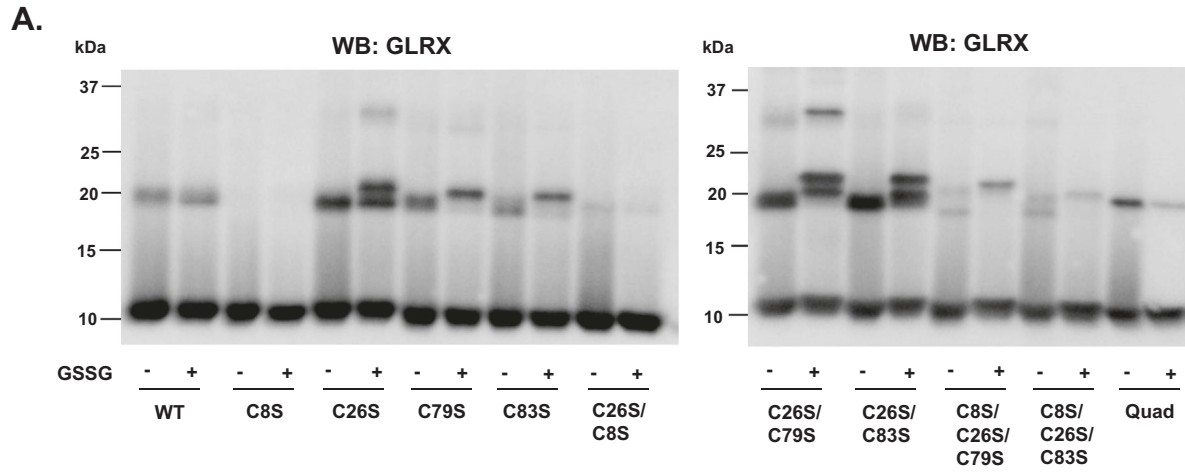

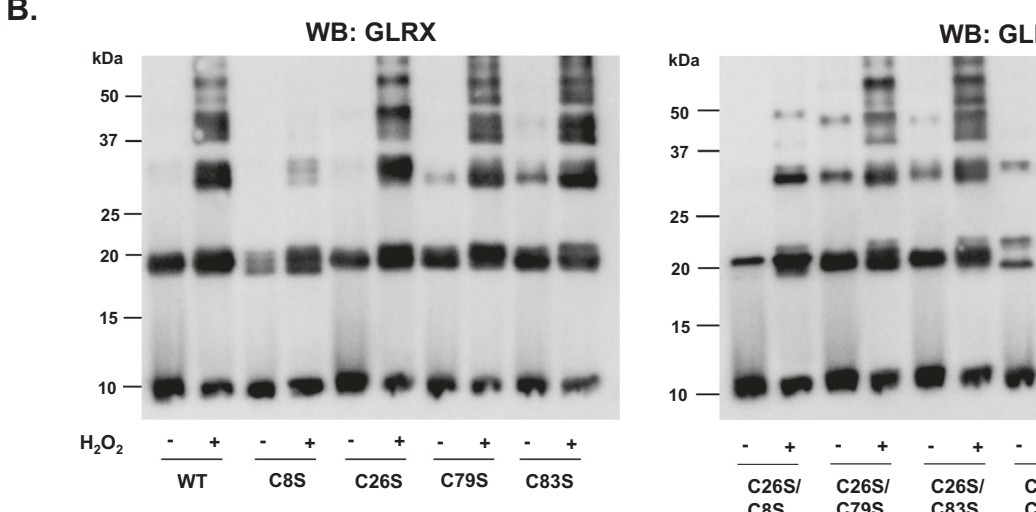

**Fig. 7 | C8 contributes to dimer formation in response to GLRX oxidation. A, B.** Western blot of reduced and oxidized GLRX mutants. Protein was incubated with either GSSG or $H_2O_2$, as in activity assays, and separated on a non-reducing 15% polyacrylamide gel, transferred to nitrocellulose membrane, blocked with 5% BSA, and probed for GLRX. Representative of $n = 3$ blots.

both experimentally and with molecular modeling, with C79 being less susceptible than other cysteines.

GLRX activity assays classically incorporated substrates like S-sulfocysteine or 2-hydroxyethyl disulfide, among others[46–48] with inherent limitations due to the non-physiological nature of those substrates and measurement of the oxidation of NADPH to NADP+, performed in the presence of GSH and glutathione reductase (GR), as a proxy for GLRX activity (Fig. 1A). The use of diE-GSSG or E-GS-BSA as biologically relevant substrates to assess GLRX activity in combination with GR, GSH, and NADPH addressed these limitations[35]. Herein, we adapted the latter GLRX assay by omitting GR, GSH, and NADPH from the diE-GSSG reaction and replacing these reducing components with 5uM DTT, analogous to the use of diE-GSSG plus 5uM DTT in protein disulfide isomerase activity assays, in order to overcome potential concerns of interference of GSH, NADPH, and GR with glutathionylated or $H_2O_2$-oxidized versions of GLRX. Using these revised conditions, our results demonstrate that GLRX's activity against E-GS-BSA was remarkably similar to that against diE-GSSG and required the presence of C23, the N-terminal cysteine in the catalytic fold required for GLRX catalytic activity. Moreover, the enhanced activity of the C26S mutant towards these substrates is consistent with previous reports and

demonstrates that GLRX is acting in a monothiol mechanism towards reduction of either substrate. This modified assay uniquely enabled us to address how the presence of each of GLRX's cysteines and their oxidation affected the rate of reduction of these GSSG or PSSG substrates.

There are a number of limitations to our present study. We did not experimentally validate the direct reactivities of each of GLRX's cysteines with $H_2O_2$ in a dose and time-dependent manner and whether these reactivities are affected by mutation of one or more of GLRX' cysteines. Assessment of GLRX exposed to $H_2O_2$ via mass spectrometry yielded complex spectra due to multiple oxidation species and impeded accurate assessment of oxidation states due to the additional aggregation of GLRX via inter-molecular disulfides (predicted in our modeling studies, Fig. 8). Moreover, mutation of one of GLRX's cysteines may affect oxidizability of other cysteines or amino acid residues. Prior work already demonstrated that C26 promotes disulfide formation while slowing down GLRX's activity as a deglutathionylase involving C23[20,21]. It is possible that substitution of additional GLRX cysteine residues similarly affects oxidation of other cysteines or other amino acids, which could explain the altered migration patterns under non-reducing electrophoresis conditions (Fig. 7A, B).

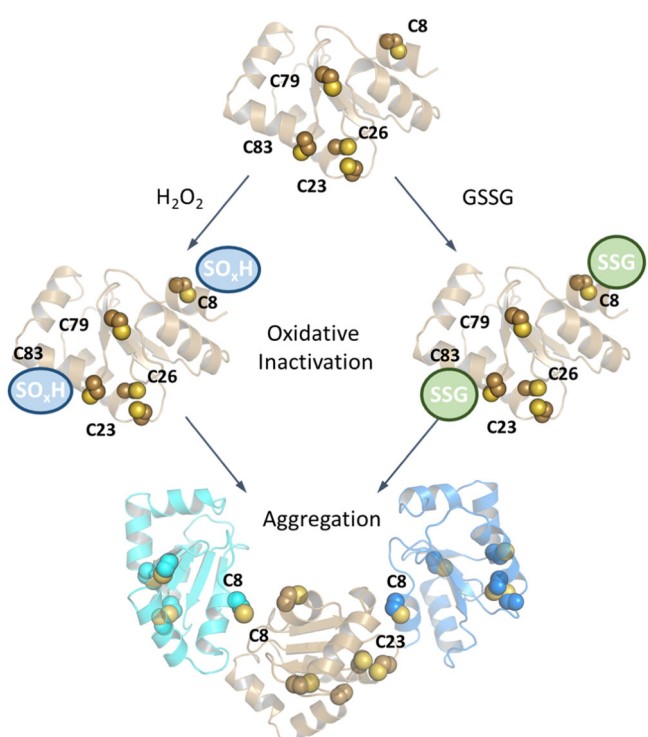

**Fig. 8 | Graphical model invoking inactivation and aggregation of GLRX by S-glutathionylation and oxidation.** Shown is the impact of GSSG or $H_2O_2$ on oxidation, highlighting oxidation of C8 and C83 and the impact of aggregation involving disulfides between C8 and C8 or C8 and C23 of different GLRX molecules.

The ability of GLRX to be glutathionylated and inactivated by its own substrate (GSH) represents a point of negative feedback regulation for the enzyme. The extent to which oxidative inactivation of GLRX through S-glutathionylation functionally contributes to the diseases in which a role of enhanced PSSG has been invoked awaits further studies. As stated earlier, the GLRX system directly depends on the GSH/GSSG redox couple, which is in turn controlled by NADPH/NADP[+] and GR. The oxidation state of these redox couples and the local activities of GR therefore can directly impact the extent to which S-glutathionylated GLRX at non-catalytic cysteine sites can be regenerated. Our findings also point to the putative importance of additional active GLRX molecules in deglutathionylation and regeneration of reduced GLRX, and raise questions about the extent to which the total pool of cellular GLRX molecules can be glutathionylated without compromising GLRX-dependent cellular redox control.

Glutathione S-transferases (GST) act as catalysts of protein S-glutathionylation, likely acting on sulfenic acid intermediates[49]. It remains unclear to date whether GST-dependent S-glutathionylation can also lead to inactivation of GLRX. This mechanism, wherein the forward catalyst blocks the reverse reaction, would provide an effective step to allow the accumulation of S-glutathionylated proteins. In a scenario wherein the oxidative burden has the potential to lead to irreversible overoxidation, maintenance of S-glutathionylation via S-glutathionylation of GLRX, until oxidative burden is alleviated, represents a potential regulatory mechanism that minimizes the need for turnover of irreversibly oxidized proteins. However, additional studies will be required to determine whether inhibition of GLRX via S-glutathionylation represents a physiologically relevant feedback mechanism.

The precise knowledge of the oxidation and glutathionylation of GLRX at the molecular level provides the basis to improve the stability and activity of the protein. The QM/MM approach used here provides insight to which cysteines are most susceptible to oxidation, critical for the design and optimization of the enzyme either through direct mutation of those cysteines that were described herein, mutation of amino acids in close proximity to lower the oxidizability of critical cysteines or the design of small molecules to protect GLRX from oxidative inactivation. Taken together, the present data indicate that mutation of C8, C26, and C83 creates a form of GLRX that is most active and resistant to oxidative inactivation and represents an attractive candidate for testing as a therapeutic in settings of dysregulated glutathionylation. However, the enhanced aggregation of C8S C26S and C83S triple mutant GLRX protein observed in response to $H_2O_2$ (Fig. 7B) will require additional structural and functional characterization to address other oxidation targets and potential implications for protein folding, turnover, or other biological features. Treatment of mice with exogenous WT GLRX has been previously reported to reverse existing fibrotic remodeling in the lung using mouse models of fibrosis[8] while treatment with GLRX2 reduces features of allergic airway disease[50]. The use of the GLRX mutants described above may be more effective in treating disease phenotypes, although further studies will be essential to formally assess the biological activity of the compound mutants of GLRX described herein.

## Methods

### Materials and reagents
The following reagents were purchased from Sigma-Aldrich; reduced glutathione (GSH), oxidized glutathione (GSSG), potassium phosphate, fatty acid free bovine serum albumin (BSA), Tris-HCl, Hank's balanced salt solution (HBSS), dithiothreitol (DTT), 0.5 M ethylenediaminetetraacetic acid (EDTA), iodoacetamide, and catalase from bovine liver. Amicon Ultra Centrifugal Filters and D-tube Dialyzer with 12–14 kDa MWCO were purchased from Millipore (Burlington MA, USA). The Detergent Compatible (DC) Protein Assay was purchased from Biorad (Hercules CA, USA) and site-directed mutagenesis kits were from Agilent (Santa Clara, CA). Di-eosine-di-glutathione (DiE-GSSG) was purchased from Cayman Chemical (Item no. 11547, Ann Arbor, MI, USA).

### Generation of GLRX Cys-Ser mutant proteins
Single cysteine to serine mutations in mouse wild type (WT) GLRX plasmid (PGEX) sequences were made using the QuikChange II XL Site Directed Mutagenesis Kit with mutation-specific primers (Supplementary Table 1) designed using Agilent's online primer design program. Mutated plasmids were produced from WT GLRX according to the manufacturer's directions and Sanger sequencing was performed using an AB 7500 Fast Sequence Detection System (Applied Biosystems, Foster City, CA). Mutant gene and protein sequences were verified by using NCBI BLAST and ExPASy Translate online tools. (Supplementary Table 2). BL21(DE3) competent cells (Agilent) were transformed with GLRX plasmids according to the manufacturer's directions. GST-tagged WT and mutant GLRX protein were purified, subjected to GST-tag and endotoxin removal using routine procedures. Final protein preparations were filter sterilized, and aliquots stored at −20 °C (See supplementary methods). Protein concentration of aliquots was determined by DC Protein Assay (Biorad) and purity assessed by Coomassie Blue gels.

### GLRX activity assay
Activity of GLRX was determined utilizing diE-GSSG as a substrate at room temperature according to ref. 35 with the following modifications. 25 ng of WT GLRX or GLRX containing Cys-Ser mutations was loaded onto a 96-well plate in a potassium phosphate buffer (50 mM $K_2HPO_4$, 1 mM EDTA, pH 7.1) containing 5 µM DTT as the reductant for GLRX analogous to its use in protein disulfide isomerase activity assays[36,51]. 150 nM of diE-GSSG was added to start the reaction. Fluorescence was immediately monitored at excitation 485/20 and emission 525/20 at 1 min intervals for 30 min using a BioTek Synergy (Winooski,

VT) plate reader. Mean activity was calculated by dividing the slope from the linear portion of each curve by GLRX protein amount in each well. Activities of mutant GLRX proteins were expressed as fold change from WT GLRX. Kinetic parameters (Km, Vmax) were calculated by varying the substrate concentration and fitting to a Michaelis–Menten curve in Kaleidagraph (5.01).

To measure the activity of each GLRX mutant towards a PSSG substrate, we used diE-GSSG conjugated to bovine serum albumin (BSA) (E-GS-BSA)[35]. BSA was first reduced with 1 mM DTT at room temperature for 30 min, and excess DTT removed with Amicon 10 kDa MWCO ultracentrifugation filters. BSA (0.1 mM) was then incubated with diE-GSSG (0.3 mM) at 37 °C for 1 hr. Excess di-E-GS and unreacted diE-GSSG were removed with Amicon 10 kDa MWCO ultracentrifugation filters, prior to dialysis of E-GS-BSA against 100 mM potassium phosphate buffer (pH 7.5) overnight. The amount of synthesized E-GS-BSA was determined using the DC Protein Assay (Biorad). GLRX activity was measured and analyzed as described above.

### Glutathionylation of WT and GLRX Cys-Ser Mutant proteins

Glutathionylation of GLRX was performed by incubation with GSSG, a commonly used method to induce glutathionylation in vitro[8,23]. GLRX proteins were first reduced with 1 mM DTT at room temperature for 30 min, followed by removal of DTT using Amicon 10 kDa MWCO ultracentrifugation filters. 2 µg of GLRX was incubated with 1 mM or 50 mM GSSG at 37 °C for 30 min. In select samples, 100 mM DTT was added for an additional 10 min incubation at 37 °C. Excess GSSG and DTT were removed through ultracentrifugation. Glutathionylated mutants were then tested in the GLRX activity assay using diE-GSSG as a substrate as described above.

### Oxidation of WT and GLRX Cys-Ser mutant proteins

Oxidation of GLRX was performed by incubation with $H_2O_2$. GLRX was reduced with DTT as described above. 2 µg of reduced protein was then incubated with 100 µM $H_2O_2$ for 10 min at 37 °C. Excess $H_2O_2$ was removed from each sample with catalase (100 units/sample) for 10 min at room temperature. Activity of GLRX was then measured using the diE-GSSG activity assay as described above.

### Gel electrophoresis and western blotting

GLRX mutants were separated by SDS-PAGE on a 15% polyacrylamide gel. Gels were stained with Coomassie blue solution for 30 min at room temperature and then destained overnight using 10% methanol and 7% acetic acid and imaged. Alternatively, proteins were transferred to a nitrocellulose membrane. Membranes were blocked with 5% BSA for 1 h and probed for GLRX overnight at 4 °C using an affinity-purified rabbit polyclonal antibody (dilution 1:1000) produced by Rockland Immunochemicals (Limerick, PA, USA) against recombinant murine GLRX, which was validated in *Glrx* knockout mouse lung tissue (Supplementary Fig. 11).

### Trypsin digestion for mass spectrometry

For confirmation of Cys to Ser mutants, WT GLRX or mutant GLRX proteins were briefly subjected to SDS-PAGE and gels were stained with Coomassie blue followed by destaining with 50% methanol and 5% acetic acid. Each sample band was excised from the gel and destaining continued until the bands were clear. Samples were then sequentially reduced with 10 mM DTT and alkylated with 100 mM iodoacetamide, each for 30 min at room temperature. Samples were then digested with 20 ng/mL trypsin at 37 °C for 16 h and the peptides were extracted from the gel for MS/MS analysis.

To identify sites of glutathionylation, WT GLRX was glutathionylated in vitro by incubation with GSSG (1 or 50 mM) prior to gel electrophoresis under non-reducing conditions. Peptides were prepared in the same way as GLRX mutants, with the exception of the DTT incubation.

### Liquid chromatography–mass spectrometry

The tryptic peptides resuspended in 2.5% $CH_3CN$ and 2.5% formic acid (FA) in water were analyzed on the Q-Exactive Plus mass spectrometer coupled to an EASY-nLC 1200 system (Thermo Fisher Scientific), as previously described (See Supplementary Information). Briefly, samples were loaded onto a 100 µm i.d. capillary column (ended with a laser-pulled orifice) that was packed with UChrom C18 (1.8 µm particle size, 120 Å, Cat. No: PN-80001; Nanolcms, CA) at a flow rate of 300 nl min$^{-1}$. Peptides from multiple mutant samples were separated by a solvent system composed of solvent A: 100% water/0.1% Formic acid (FA) and solvent B: 80% $CH_3CN$/0.1% FA with a gradient of 0-44% B over 60 min (for GLRX samples with multiple mutations) 0-44% B over 150 min (for glutathionylated GLRX and single mutation GLRX samples), 44-100% B in 1 min and then 100% B for 8 min, followed by an immediate return to 100% A and a hold at 100% A for 20 min before the next injection. Mass spectrometry data were acquired with data-dependent "Top 10" acquisition. WT or GLRX mutant protein samples were randomized in run order and peptide standards (bovine serum albumin) were run between samples. The masses of the peptides that were identified in both replicates by data-dependent acquisition mentioned above were imported into the inclusion list of the parallel reaction monitoring (PRM) workflow. The closely eluting VVVFIKPTC(GSH)PYC(O3)R and VVVFIKPTC(O3)PYC(GSH)R peptides were distinguished and quantified by PRM. PRM was carried out with alternating MS-SIM and PRM scans (2 scan groups) using a shorter 60-min of 0−44% B gradient. Full scans were acquired from $m/z$ 300-2,000 at 70, 000 resolution (AGC target 1e6; max IT 100 ms; profile mode). PRM were carried out with higher-energy collisional dissociation (HCD) MS/MS scans in profile mode at 17,500 resolution on the precursors of interest, with the following settings: AGC target 5e4; max IT 100 ms; isolation width of 1.6 m/z and a normalized collisional energy of 26%. Lock mass function was activated (m/z 371.1012; use lock masses: best; lock mass injection: full MS).

### Database searches

Raw files were analyzed using the Proteome Discoverer 2.4 (Thermo Fisher Scientific) and product ion spectra searched using the SEQUEST with the "Basic" Processing and Consensus workflows against a Uniprot *Mus musculus* protein database (UP000000589). Search parameters included full trypsin enzymatic activity; max two missed cleavages, mass tolerance at 10 ppm, and 0.02 Da for precursor ions and fragment ions. Global dynamic modifications included: methionine oxidation (+15.995 Da), cysteine carbamidomethylation (+57.021 Da), oxidation (+15.995 Da), dioxidation (+31.990 Da), and trioxdiation (+47.985 Da), Cys->Ser (−15.977 Da (C) and S-glutathionylation (+305.068 Da). Dynamic Modification specific to the peptide terminus included: N-terminus acetylation (+42.011 Da), Met-loss (−131.040 Da), and Met-loss+Acetyl (−89.030 Da). PSM validation was completed with the Fixed Value PSM Validator. The "IMP-ptmRS", "Peptide Isoform Grouper", and "Modification Sites" nodes were included in the workflow for distinguishing peptides with multiple modification sites within the same peptide from glutathionylated GLRX samples.

### Data analysis

Mass spectra were manually inspected using Scaffold Q + S 4.11 (Proteome Software, OR) to confirm the mutation sites and glutathionylation sites in GLRX. The corresponding spectra with the highest XCorr value are included in the Supplementary Figs. 1 and 2 (mutation sites) and 3 (glutathionylation sites). For the quantification of glutathionylated peptides (two independent experiments), the search files (.msf) together with the raw files were imported into the Skyline for selecting the precursors or transitions for quantitation (Supplementary methods). Abundance of the GLRX peptides was determined by constructing the extracted ion chromatograms (XICs) and determining the areas

under the Satvisky Golay-smoothed ion elution profiles. Additional quantification by PRM were performed on closely eluting VVVFIKPTC(GSH)PYC(O3)R and VVVFIKPTC(O3)PYC(GSH)R, when necessary. Boundaries of integration were manually evaluated according to their retention times. For the closely eluting VVVFIKPTC(GSH)PYC(O3)R and VVVFIKPTC(O3)PYC(GSH)R peptides, the abundance was determined by – summing the area of the XIC of the 3 transitions that are unique to the peptides (VVVFIKPTC(GSSG) PYC(O3)R: $m/z$ 939.433 ++ → 326.113 [y2]$^+$,C(GSH), 489.176 [y3]$^+$ Y, and 586.229 [y4]$^+$ P, and VVVFIKPTC(O3)PYC(GSH)R: ++ → 538.196 [y2]$^+$C(O3), 746.260 [y3]$^+$ Y, and 843.312 [y4]$^+$ P) (diagnostic ions are boxed in the ion tables in Supplementary Fig. 3). The chromatographic traces were exported from Skyline to GraphPad Prism 8 for chromatogram plotting. The spectra with the highest XCorr acquired during peak elution are included in the Supplementary Fig. 3.

### Molecular dynamics simulations of GLRX and complexes
Using the crystal structure of GLRX (PDB ID: 4RQR, Uniprot ID: P35754), we constructed models of GLRX wild type and mutants (C83S, C8S, and C79S) as well as the GLRX dimers for molecular dynamics (MD) simulations, using Protein Preparation Wizard (Pymol, Schrödinger, Inc.) and the relaxation strategy in our prior work[52]. The simulation box was built with the SPC water model and the OPL3e force field[53]. All the NPT simulations (300 K, 1 atm) were carried out in the *Desmond* program (Schrödinger, LLC) on graphics processing units (GPUs), with a recording interval of 9.6 ps and the van der Waals and short-range electrostatics cut off at 9 Å. Each GLRX wild type or mutant has two 400-ns simulation replicas while each GLRX multimer was simulated with two replicas, 200 ns each (Supplementary Table 3). The multimer conformations were obtained from alignment to a dimer crystal structure (PDB ID: 3UIW) or self-assembly of free monomers. All these simulations were analyzed using the Simulation Event Analysis tool implemented in *Maestro*.

### $pK_a$ calculation of GLRX cysteines
The $pK_a$ calculation was performed with the *PROPKA* package[54,55] available in the *Maestro* program. The $pK_a$ calculation was carried out with the equilibrated protein (described above) and any water molecule further than 5 Å from the protein was removed to diminish the calculation time.

### QM/MM Calculations of GLRX oxidation
To gain insight into the oxidation of cysteine residues in the presence of $H_2O_2$ at the molecular level, QM/MM calculations were performed using the *Qsite* software package[56,57]. We used the equilibrated protein models from the MD simulations as the input structures for the QM/MM optimizations. Each cysteine residue, two adjacent residues in the sequence, and a $H_2O_2$ molecule were included in the QM region. The rest of the protein was treated as the MM region. The structures were first optimized at the B3LYP/6-31 G* level of theory, with a non-bonded cutoff of 12 Å. Next, all the final energies were obtained from single point energy calculations at B3LYP/6-31 G** level with the PBF continuum solvation model for water implemented in *QSite*. Harmonic zero-point energy corrections were included for the calculation of the activation energies, as well as for the reaction enthalpies (Supplementary Table 5), and the reaction Gibbs free energies (Supplementary Table 5).

We first generated initial guesses for all the transition state coordinates, informed by the distances of a model transition state [CH₃S···HOOH]⁻. We used the oxidation of methanethiol with hydrogen peroxide as the model reaction, based on extensive prior investigation[37,58] and the well-established reaction pathway. The thiol is deprotonated first, before reacting with $H_2O_2$, with the rate-determining transition state of the reaction being methanethiolate attack on $H_2O_2$. Therefore, we optimized the model transition state for

methanethiolate oxidation at the B3LYP/6-31 G* level with the *Jaguar* (version 10.4) software package[59] and verified the first-order saddle point nature of the transition state with a frequency calculation at the B3LYP/6-31 G* level. The optimized transition state (see Supplementary Information for the coordinates) displayed an imaginary frequency of 263 cm⁻¹. Next, we further confirmed that the transition state connects the desired reactants and products with an intrinsic reaction coordinate (IRC) calculation (see Supplementary Fig. 4). The optimized [CH₃S···HOOH]⁻ model transition state complex displayed two key distances, characteristic of the transition: (i) The S···O distance (2.2 Å) of the forming S-O bond and (ii) The O···O distance (1.9 Å) of the breaking O-O bond. To obtain good initial guesses for the QM/MM transition state optimizations, we constrained these two key distances to the values found for the model [CH₃S···HOOH]⁻ transition state complex. We then performed energy minimizations with these distance constraints, and finally performed full transition state optimizations of the resulting structures without any constraints. The transition state searches were carried out using B3LYP/6-31 G** for C79, C26, and C83 and B3LYP/6-31 G*⁺ for C23 and C8, as this protocol reliably locates transition states (verified by a single imaginary frequency obtained for each transition state − see Supplementary Table 9).

### Induced-fit docking of GSSG to GLRX
All the protein-ligand docking was performed with the *Induced Fit Docking* (IFD) package[60] with the GSSG molecule minimized in solvent using DFT calculations (B3LYP-6-31G**) and QM-calculated partial charges. The receptor conformation was taken from the final snapshot of a 200-ns GLRX simulation. GSSG was docked to the proximity of all five cysteine residues, with each binding site defined as the center of the cysteine ($i$) and adjacent residues ($i \pm 1$). The inner and outer box sizes were set as 10 and 25 Å respectively (except 30 Å for the outer box of C23). In the IFD workflow, molecular docking in *Glide* was first performed, followed by side-chain and loop refinement within 5 Å of GSSG in *Prime*[61–65]. Finally, GSSG was redocked to the refined protein conformations and the Glide XP score was calculated to select the top poses.

### Statistical analyses
All studies were conducted three times, with the exception of the mass spectrometry studies which were conducted twice. Statistical analyses were performed using Prism software version 9.0.1 (GraphPad, San Diego, CA). GLRX activity values were compared using a one-way ANOVA with Dunnett's multiple comparisons, with each mutant compared to the activity of the WT protein. Statistical analysis of mutant inactivation was performed using a t-test between the reduced activity of the unmodified enzyme, normalized as 100%, and the percentage of reduced activity measured following incubation with GSSG or $H_2O_2$. P values reported on graphs as follows: *$p \le 0.05$, **$p \le 0.01$, ***$p \le 0.001$, ****$p \le 0.0001$.

### Reporting summary
Further information on research design is available in the Nature Portfolio Reporting Summary linked to this article.

## Data availability
The mass spectrometry proteomics data have been deposited to the ProteomeXchange Consortium via the PRIDE partner repository with the project identifier PXD026486. PDB entries 4RQR and 3UIW were used for modeling. Uncropped bolts and gels are provided in Supplementary Data 1. Source data are provided with this paper.

## Code availability
Access to code used in these studies is available upon request.

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

## Acknowledgements
This work was supported by grants NIH R35HL135828 (Y.-J.H.), NIH R01HL137268, R01HL085646, R01HL138708, R21AG055325 (Av.dV.), NIH R01HL122383 and R01HL141364, R01HL136917 (V.A.). S.T.S. was supported by the ARO through a Young Investigator Award (Grant #71015-CH-YIP) and an NIH R35 award (R35-GM147579). E.M.C. was supported by NIH T32 HL076122 and a Parker B. Francis Fellowship for Pulmonary research. Y.W.L., C.G., and the proteomics facility are supported by NIH P20GM103449 (Vermont INBRE: Vermont Biomedical Research Network). We thank Sydney Cohn Guthrie from the Vermont Biomedical Research Network Proteomics Facility and Dr. Cristina Furdui (Wake Forest University) for assistance and advice with Mass Spectrometry assessments.

## Author contributions
Laboratory experiments and data collection were performed by E.M.C., with experimental assistance from M.M. and S.W. and guidance from Y.M.W.J.H.; Y.W.L., C.G., and A.M.M. helped with mass spectrometry experiments and data analysis. Molecular modeling was performed by M.S., S.T.S., and J.L. Y.M.W.J.H. and E.M.C. wrote the manuscript. R.H., A.V., and V.A. edited the manuscript and assisted with analysis of results.

## Competing interests
Yvonne Janssen-Heininger and Vikas Anathy hold patents: United States Patent No. 8,679,811, "Treatments Involving Glutaredoxins and Similar Agents" (YJ-H, VA), United States Patent No. 8,877,447, "Detection of Glutathionylated Proteins" (YJ-H), United States Patents 9,907,828 and 10,688,150 "Treatments of oxidative stress conditions" (YJ-H, VA). Yvonne Janssen-Heininger and Vikas Anathy have received consulting fees from Celdara Medical LLC for their contributions to the proposed commercialization of glutaredoxin for the treatment of pulmonary fibrosis. The remaining authors declare no competing interests.
