## [Peer Review File · Nature Communications]

REVIEWER COMMENTS

Reviewer #1 (Remarks to the Author):

The paper "STRUCTURAL AND FUNCTIONAL FINE MAPPING OF CYSTEINES IN MAMMALIAN GLUTAREDOXIN REVEAL A HIERARCHY OF SUSCEPTIBILITY TO OXIDATIVE INACTIVATION" by Corteselli et al. addresses the potential regulatory role of cysteine side chain redox modifications for the oxidoreductase activity of mammalian glutaredoxin 1. Using an interdisciplinary computational and biochemical approach, the authors have mapped redox modifications induced by high concentrations of GSSG and H₂O₂ in equilibrium reactions, the susceptibility of the cysteine side chains to undergo the detected redox modifications were analyzed by QM/MM calculations and simulations. The topic of protein S-glutathionylation and its regulation in vivo is relevant for a wide range of physiological and pathological conditions and therefore of potential interest to a wide audience. The work does confirm and expands previous studies. However, the manuscript, as presented, suffers from a severe lack of control experiments and details, as well as insufficient presentation of the data such as the lack of absolute activity data.

Major points

1. The manuscript was written in a puzzling way. The authors repeatedly state that glutaredoxins are specific for the catalysis of "reversible deglutathionylation", frequently expressed in a way that implies that glutaredoxins catalyze the reaction in one way only, i.e. de-glutathionylation. If this was intended, it was based on a fundamental misconception. Glutaredoxins catalyze the reaction in both directions dependent on the thermodynamic constrains. This was best demonstrated by the introduction of roGFP-based redox sensors in various organisms, that respond to changes in the intracellular glutathione redox dynamics. Oxidation and reduction in vivo occur through a glutaredoxin-catalyzed S-(de-)glutathionylation step, both directions depend on the presence of a glutaredoxin (see for instance Meyer et al., *Plant Journal*, 52:973-86; 2007 and Gutscher et al. *Nature Methods*, 5:553-9; 2008). Moreover, mouse embryonic fibroblasts from GLRX1 knock out animals do not show an increase in protein S-glutathionylation pattern per se (Ho et al., *Free Radic Biol Med* 43:1299-312; 2007). This is even more puzzling, when considering that one of the major assays applied by the authors to study Grx activity "determined utilizing diE-GSSG as a substrate" is the first half reaction of the (forward) glutathionylation reaction.
2. Most of the oxidative modifications described in this study, as well as some of the effects on activity, have been reported before, mostly in the cited study by Hashemy et al. (*J. Biol. Chem.* 282:14428-14436; 2007). The novelty of the paper is therefore largely limited to the individual susceptibilities of the cysteine side chains to oxidative modifications. These are based on theoretical calculations and were not confirmed experimentally, e.g. by titration experiments. Moreover, as of today, it is not clear which of these modifications might occur or be relevant under physiological or pathological conditions or in cellular models thereof. In this respect, a clear weakness of the study is the use of very high concentrations of GSSG (50 mM) for the induction of S-glutathionylation in vitro, way beyond any physiological/pathological meaningful concentration (in the μ M range only).
3. Kinetic analyses – The authors use nM concentrations of labeled GSSG and μ M concentrations of GSH in their activity assays. The authors express the activity in relative/normalized form only, calculated as the linear part of the initial rate divided by protein concentration, i.e. at one substrate and one enzyme concentration only. The Michaelis constant of human Grx for GSH is in the range of 2 mM, hence the concentration of glutathione in all assays was

strongly limiting. It is therefore not clear whether the reported modifications effect the activity of the protein, or its affinity for its substrates.

- All data must be expressed in absolute values as well, to allow comparisons with previous studies and to judge the relevance of the data reported here.
- Proper controls must be provided that demonstrates the linear dependence of the activity on the enzyme concentration in the assay.
- To fully evaluate the effect of the redox modifications, kcat and Km values must be determined separately.
- Suitable controls of the stability of the redox modifications are lacking as well. The authors should provide a time dependence of the modifications' presence, especially under assay conditions.
- [Fig 5C] Where does the second band (~20kDa) in reduced samples come from? WT red and WT GSSG look identical! So the decreased activity in fig 5B can not be due to the dimer formation? Likely, reduction was incomplete, putting much of the results and conclusions in doubt.

Statistics – the use of the SEM when comparing two data sets for potential significant differences is meaningless. The data presented here is not compared to a total population, SD is the appropriate parameter. The authors also only state $n \geq 3$. The authors should provide the exact number for all assays and meaningful tests that these data sizes are sufficient.

Specific points

180 No controls were provided that the presence of catalase does not influence the activity measurements (catalase can undergo glutathionylation and is a potential substrate!)

269 Were 200ns sufficient for the MD of the dimer? No rmsd of these simulations or other controls were provided.

338 Are the rmsd of thee MD sufficient to conform the stability of the structure? Does the radius of gyration change over the time frame of 400ns? Could the stability of the mutants be also confirmed in vitro (see also above)?

340 [S.Fig 1] The rmsd in supp fig 1 only depicts one of the replica.

[Fig 2 B+C and E+F] Figures 2C and 2F basically repeat the information shown in fig 2B and 2E, respectively, won't one of each would be sufficient? Similar for the figures 3B+C and 3E+F.

397 It is not clear how the simulation box with 3 free Grx monomers was build: What was the initial distance between the monomers? Which conformations were they in?

399 The MD simulations only demonstrate the importance of C8-C8 and C23-C8 interactions for the formation of the dimer, what about C83? And C79 (see the Hashemy et al. paper mentioned above)

[Fig 4D] is hard to understand {quality of the print?} only two monomers/two different colors are visible in the fig. Maybe also include a zoomed out figure of the structure as well? To get a general idea about the structure of the dimers?

417-420 On what is the conclusion of the "slightly different dynamics" based, especially since the comparison of the abundance is not possible?

427 [Fig 5B] Only C8 and C83 mutants mentioned as those that remained their activity after the incubation with GSSG. what about C26? the reduction of the activity was either not significant or the significance was not included in the figure.

476-477 [Fig 7C] Why do the triple mutants with C8 (C8/26/79S and C8/26/83S) show no reduction of oligomerisation as the single and quad mutant?

[Fig 7C] It seems like the ~20kDa band consist out of two bands. The C8/26/79S mutant shows clearly two bands: 20 and ~22 kDa. Where does the second band come from? (see also Fig. 5)

Reviewer #2 (Remarks to the Author):

The manuscript of Elizabeth Corteselli et al. is unfortunately very disappointing and additional experiments are needed before it potentially could be accepted for publication. Further the way that the study it is currently written is rather basic, as the authors present a collection of observations and then the discussion does not really go deeper.

One major point is the physiological significance of all the experiments. Authors are really using huge GSSG concentrations that will never occur in the cytosol - can you really speak about "hierarchies" then? Can glutathionylation really be protective as they suggest as GSH concentrations remain high after "oxidative insults" (see previous studies)? No mention of GR and kinetic competition with it. Basically, a lot of critical points...

Suggestion: Authors might have a look at the review of Marcel Deponte 2017.

How to improve the manuscript: A thorough comparative kinetic stopped flow analysis with GR could be a good starting point.

REVIEWER COMMENTS

Reviewer #1

Major points

Comment 1. The manuscript was written in a puzzling way. The authors repeatedly state that glutaredoxins are specific for the catalysis of “reversible deglutathionylation”, frequently expressed in a way that implies that glutaredoxins catalyze the reaction in one way only, i.e. deglutathionylation. If this was intended, it was based on a fundamental misconception. Glutaredoxins catalyze the reaction in both directions dependent on the thermodynamic constraints. This was best demonstrated by the introduction of roGFP-based redox sensors in various organisms, that respond to changes in the intracellular glutathione redox dynamics. Oxidation and reduction *in vivo* occur through a glutaredoxin-catalyzed S-(de-)glutathionylation step, both directions depend on the presence of a glutaredoxin (see for instance Meyer et al., *Plant Journal*, 52:973-86; 2007 and Gutscher et al. *Nature Methods*, 5:553-9; 2008). Moreover, mouse embryonic fibroblasts from GLRX1 knock out animals do not show an increase in protein S-glutathionylation pattern *per se* (Ho et al., *Free Radic Biol Med* 43:1299-312; 2007). This is even more puzzling, when considering that one of the major assays applied by the authors to study Grx activity “determined utilizing diE-GSSG as a substrate” is the first half reaction of the (forward) glutathionylation reaction.

Response 1: We thank the reviewer for their interest and pointing out key aspects of glutaredoxin catalysis. We are well aware that glutaredoxins can catalyze both deglutathionylation and glutathionylation reactions. We apologize to the reviewer for the confusion we caused with the terminology and have corrected this in the revised manuscript. The scope of the present study was to focus on the deglutathionylation reaction because, as the reviewer pointed out, this is the favorable reaction under physiological conditions wherein the GSH/GSSG redox couple is highly reduced (PMID: 14713336)

The first step measured in our activity assay utilizing di-E-GSSG involves the reduction of glutathione disulfide by GLRX (PMID: 26836485). The reviewer is correct in noting that GLRX itself is glutathionylated as a result of this reaction, but the activity the enzyme exerts on the substrate is that of substrate reduction/deglutathionylation. To further corroborate this point, we also utilized the substrate E-GS-BSA to model deglutathionylation of a protein target. Again, the first step of the reaction of GLRX with diE-GS-BSA is the removal of E-GS by GLRX, yielding BSA-SH, E-GSH and GLRX-Cys23-SSG. In this case as well, GLRX is performing deglutathionylation on the BSA molecule. In order to clarify this we have created a new Figure (1D) that delineates these reactions.

Numerous studies by our laboratory have demonstrated increases in S-glutathionylation in the absence of *Glrx* and further increases in response to a number of (patho)physiological stimuli, consistent with mammalian *Glrx*' role as a deglutathionylase under physiological conditions. In response to the reviewer's comment regarding glutaredoxin knockout embryonic fibroblasts, referencing the Ho manuscript, those authors did not quantify S-glutathionylation in WT or *Glrx*^{-/-} MEF, instead shown are 2 lanes on a Western blot (PMID: 17893043) that are not quantitative. We (in collaboration with Dr. Ho) and others have published extensively that lungs from mice lacking *Glrx* and *Glrx*^{-/-} airway epithelial cells have increased levels of S-glutathionylation, both at baseline and when exposed to various stimuli and insults (PMID: 33817836, PMID: 32971362, PMID: 29988126, PMID: 27035878, PMID: 16515838, PMID: 34876574, PMID: 16916935). These observations collectively point to the function of mammalian GLRX under physiological conditions being that of a deglutathionylase.

Comment 2: Most of the oxidative modifications described in this study, as well as some of the effects on activity, have been reported before, mostly in the cited study by Hashemy et al. (J. Biol. Chem. 282:14428-14436; 2007). The novelty of the paper is therefore largely limited to the individual susceptibilities of the cysteine side chains to oxidative modifications. These are based on theoretical calculations and were not confirmed experimentally, e.g. by titration experiments. Moreover, as of today, it is not clear which of these modifications might occur or be relevant under physiological or pathological conditions or in cellular models thereof. In this respect, a clear weakness of the study is the use of very high concentrations of GSSG (50 mM) for the induction of S-glutathionylation in vitro, way beyond any physiological/pathological meaningful concentration (in the μM range only).

Response 2: The reviewer correctly states that this study builds upon the work reported by Hashemy et al, which examined the oxidative inactivation of GLRX using the HED assay (PMID: 17355958). Indeed, that study formed the scientific premise for conducting the present study. The Hashemy manuscript speculated about which cysteines are involved in oxidative inactivation of GLRX by GSSG and H_2O_2 . However, they did not provide in silico nor experimental evidence of their involvement through the demonstration of improved activity following point-by-point mutations of each of the cysteines or in combination. Furthermore, the Hashemy manuscript utilized a GLRX activity assay that does not specifically model deglutathionylation of a protein, and relied on the concurrent activity of glutathione reductase. The main contribution of our manuscript is the elucidation of the role that each individual cysteine plays in inactivation by glutathionylation and oxidation, and in aggregation, as measured by a more specific and physiologically relevant enzyme assay that incorporates the deglutathionylation of a physiologically relevant substrate.

More importantly, in tandem to the experimental studies, we conducted sophisticated quantum mechanics and molecular modeling studies, not conducted before that provided novel in silico insights about the reactivities of each of GLRX's cysteines towards H_2O_2 or GSSG, or aggregation of GLRX, studies that were validated experimentally. The reviewer correctly notes that our conclusions are based on molecular modeling and experimental validation. In the present study, we applied for the first time a combined strategy consisting of molecular simulations and hybrid QM/MM calculations to gain molecular understanding of the structure-function relationships of all GLRX cysteine residues. In particular, we placed emphasis on delineating the impact of H_2O_2 -linked oxidation of GLRX by calculating the reaction thermodynamical activation parameters for all the cysteine residues present. Through mutation of GLRX cysteines, alone and in tandem, we determined their involvement in GLRX inactivation. The aforementioned Hashemy study did not create cysteine mutants of GLRX and did not provide structural information about the vulnerability of each of GLRX's cysteines towards their oxidation. This information is now provided in the revised manuscript and in our opinion presents a substantial advancement of knowledge.

The reviewer is also correct in noting that we use a supraphysiological concentration of GSSG in order to induce glutathionylation of GLRX. This concentration is in agreement with what others have previously reported to inactivate GLRX, including Hashemy et al (PMID: 17355958, PMID: 29988126). The oxidant signals that induce S-glutathionylation in vivo remain unclear and are highly likely to be context and organ dependent. For example in the lung, glutathione S-transferase P is a key contributor to protein S-glutathionylation (PMID: 27358914, PMID: 34624602) possibly involving a short lived sulfenic acid or sulfenylamide intermediate. The scenarios are impossible to precisely mimic in the test tube. We explicitly do not claim that GSSG-induced S-glutathionylation of GLRX would occur physiologically, but that this instead is the most direct chemical method to induce glutathionylation in the test tube, validated by Mass Spectrometry data provided in Figure 4A, B and Supplementary Figure 3. In the revised manuscript we have clarified the explanation of our dose of GSSG used.

Comment 3: Kinetic analyses: The authors use nM concentrations of labeled GSSG and μM concentrations of GSH in their activity assays. The authors express the activity in relative/normalized form only, calculated as the linear part of the initial rate divided by protein concentration, i.e. at one substrate and one enzyme concentration only. The Michaelis constant of human Grx for GSH is in the range of 2 mM, hence the concentration of glutathione in all assays was strongly limiting. It is therefore not clear whether the reported modifications effect the activity of the protein, or its affinity for its substrates.

Response 3: We apologize for the confusion. The reviewer is correct in noting that we utilized nM concentrations of diE-GSSG or diE-GS-BSA substrates. In this manuscript, we adapted the diE-GSSG or diE-GS-BSA GLRX assay by omitting GR, GSH and NADPH from the diE-GSSG reaction and replacing these reducing components with a minimal concentration (5 μM) of DTT, which is not sufficient to reduce diE-GSSG, analogous to the use of diE-GSSG plus 5 μM DTT in protein disulfide isomerase activity assays (PMID: 30735910, PMID: 17561094). We made this modification in order to overcome potential concerns of interference of excess GSH, NADPH and GR with glutathionylated or H_2O_2 -oxidized versions of GLRX. Using these revised conditions, our results demonstrate that GLRX's activity against E-GS-BSA was remarkably similar to that against diE-GSSG and required the presence of cysteine 23, the N-terminal cysteine in the active site known to be required for GLRX catalytic activity. Moreover, the enhanced activity of the C26S mutant towards these substrates is consistent with previous reports and demonstrates that GLRX is acting in a monothiol mechanism towards reduction of either substrate. This modified assay uniquely enabled us to address how the presence of each of GLRX's cysteines and their oxidation affected the rate of reduction of these GSSG or PSSG substrates. This clarification has now been provided in revised Figure 1D and in the discussion of the revised manuscript.

Comment 4: All data must be expressed in absolute values as well, to allow comparisons with previous studies and to judge the relevance of the data reported here.

Response 4: Given the wide varieties in GLRX preparations, instrument settings, diE-GSSG or diE-GS-BSA preparations and the inclusion of DTT used herein, we are not convinced that comparisons of absolute values will help in comparing data from previous studies. For example, in the present study all studies are conducted with murine recombinant GLRX whereas prior studies, including those mentioned by the reviewer earlier, were conducted with GLRX from different species.

Comment 5: Proper controls must be provided that demonstrates the linear dependence of the activity on the enzyme concentration in the assay.

Response 5: We agree with the reviewer that this information is pertinent. We have now included data to demonstrate the linear dependence of the activity based upon increasing concentration of GLRX present. This new data is provided in Figure 2A.

Comment 6: To fully evaluate the effect of the redox modifications, k_{cat} and K_{m} values must be determined separately.

Response 6: We have calculated K_{cat} and K_{m} values for the WT GLRX but could not provide these measurements for all the mutants following treatment with oxidants, due to the costs of additional protein purifications. This data has been added to the text of the manuscript and is shown in Figure 2A,B.

Comment 7: Suitable controls of the stability of the redox modifications are lacking as well. The authors should provide a time dependence of the modifications' presence, especially under assay conditions.

Response 7: As is shown in the western blots presented in Figures 7 F and G, which were run under non-reducing conditions 30 minutes post-treatment with H₂O₂ or GSSG, the redox modifications are still present. In consultation with Dr. Cristina Furdui (Wake Forest University) we conducted additional Mass Spectrometry studies with H₂O₂-treated GLRX which yielded various oxidized (sulfenic, sulfonic) species, along with aggregates (reflecting intermolecular disulfides) that were impossible to precisely quantify. Although this data is not included in the revised manuscript, it corroborated the presence of oxidized species under the assay conditions.

Comment 8: [Fig 5C] Where does the second band (~20kDa) in reduced samples come from? WT red and WT GSSG look identical! So the decreased activity in fig 5B can not be due to the dimer formation? Likely, reduction was incomplete, putting much of the results and conclusions in doubt.

Response 8: The reviewer is correct in noting that in this non-reducing GLRX Western blot the WT control and GSSG conditions are similar in the western blot shown in Figure 5C (now figure 6D). The 20 kDa band present in the control sample reflects the fact that our studies were conducted in room air, similar to prior observations by the group of Arne Holmgren who discovered GLRX and demonstrated that Cysteine 7 of human GLRX readily contributes to dimerization (PMID: 8549805). However, this blot merely shows that glutathionylation of WT GLRX does not enhance its propensity to aggregation beyond the background observed in the WT protein under aerobic conditions. What this blot does not capture is the amount of glutathionylation in WT GLRX following GSSG incubation, which was measured by mass spectrometry. As is shown in Figures 4 A and B, GSSG result in qualitative increases in glutathionylation of specific cysteines of WT GLRX in a dose-dependent manner, calculated from area under the curves for each of the Cysteine containing peptides.

There is a baseline amount of aggregation in the WT and several other mutants which is discussed in the manuscript and in our comments to reviewer 2 below. This is diminished by mutation of Cysteine 8 (equivalent to human Cysteine 7). However, prior to our in vitro oxidations, we incubated each mutant with dithiothreitol to reduce any oxidation that may have occurred during storage. Mass spectrometry analysis revealed under these assay conditions that WT GLRX was indeed fully reduced prior to the incubation with GSSG (not shown).

Comment 9: Statistics – the use of the SEM when comparing two data sets for potential significant differences is meaningless. The data presented here is not compared to a total population, SD is the appropriate parameter. The authors also only state n>=3. The authors should provide the exact number for all assays and meaningful tests that these data sizes are sufficient.

Response 9: We thank the reviewer for their suggestion, and have changed the statistics to standard deviation and clarified the number of replicates where appropriate.

Specific points

Comment 10: No controls were provided that the presence of catalase does not influence the activity measurements (catalase can undergo glutathionylation and is a potential substrate!)

Response 10: We apologize for not including the controls in the manuscript data. We did indeed perform a control condition in which catalase was shown to not alter the performance of the activity assay. To further clarify; the catalase was added to samples that contained H₂O₂ (with the goal of reducing excess H₂O₂), not GSSG, so there was no potential for catalase glutathionylation in those studies.

Comment 11: Were 200ns sufficient for the MD of the dimer? No rmsd of these simulations or other controls were provided.

Response 11: We thank the reviewer for this question. We have performed simulations of GLRX monomers for 400 ns each and analyzed the RMSD compared to the crystal structure (PDBID: 4RQR). The protein backbone RMSD fluctuated between 1 to 3 Å (see figure below). Such fluctuation in GLRX was consistent in our simulations presented in Supplementary Table 3, regardless the assembly state. In addition to backbone RMSD to the crystal structure, we also carried out the RMSF analysis, which agrees well with the RMSD plots. Our data show that the assembly state, the oxidation state, or single-point alanine/serine mutants does not affect the overall folded structure.

Our monomer simulations suggested that 200 ns is sufficient to capture internal fluctuation of the protein. For dimerization simulations, we had two constructs. The first construct started from a dimer structure with a symmetrical interface and the final RMSD is small (<3.5 Å) from the starting dimer conformation. The second construct started from three free GLRX molecules, and the final assembly model formed at ~150 ns and remained stable for the rest of the simulation. Therefore, in terms of intermolecular and intramolecular interactions, 200 ns is sufficient for our simulations. Also, we have two replicas for each construct, and the consistent RMSDs and RMSFs support our choice of simulation length.

Figure R1. Time evolution of backbone RMSD. The crystal structure (PDBID: 4RQR) was used as the reference structure. C23 and C26 were reduced. These simulations show a low level of structural fluctuation.

Figure R2. RMSF plots to compare reduced and oxidized wildtype and a reduced mutant in our 400-ns all-atom simulations. In the reduced wildtype, C23 and C26 are reduced while in the oxidized wildtype, C23 and C26 are crosslinked. These simulations suggest the structural rigidity of GLRX.

Comment 12: Are the rmsd of these MD sufficient to conform the stability of the structure? Does the radius of gyration change over the time frame of 400ns? Could the stability of the mutants be also confirmed in vitro (see also above)?

Response 12: Thank you for this comment. As we mention in Response 11, the RMSD measurement is sufficient to confirm the structural stability of GLRX in solution. The radius of gyration ($13 \pm 0.3 \text{ \AA}$) did not change over the course of 400 ns. The stability of the serine mutants has been experimentally confirmed, as the mutants all displayed the expected GLRX activity profiles based on previous studies (ie C23S inactive, C26S or those containing C26S more active), while other Cys-ser mutants have activities similar to WT GLRX.

Comment 13: [S.Fig 1] The rmsd in supp fig 1 only depicts one of the replica.

Response 13: This RMSD plot is just an example. All of our simulations show a backbone RMSD of 1 to 3 \AA for the protein compared to the crystal structure.

Comment 14: [Fig 2 B+C and E+F] Figures 2C and 2F basically repeat the information shown in fig 2B and 2E, respectively, won't one of each would be sufficient? Similar for the figures 3B+C and 3E+F.

Response 14: We recognize the redundancy. In the revised manuscript we have therefore consolidated the data in these figures to be more concise.

Comment 15: It is not clear how the simulation box with 3 free Grx monomers was build: What was the initial distance between the monomers? Which conformations were they in?

Response 15: The construct was built with three arbitrarily placed GLRX models (prepared from PDBID: 4RQR). In the initial model, each GLRX was at least 5 \AA from another. We have added the details in the manuscript as below

In the Methods:

The multimer conformations were obtained from alignment to a dimer crystal structure (PDB ID: 3UIW) or self-assembly of free monomers.

In the Supplemental Information:

MD Simulation of GLRX Multimers

The MD simulation of GLRX dimer (monomeric conformation PDBID: 4RQR) aligned to zebrafish GLRX (PDBID: 3UIW) represents the symmetrical C8-C8 interface (shown in surface mode on Supplementary Figure 2) which is driven mainly by polar interactions. In addition, three GLRX monomers (PDBID: 4RQR) were arbitrarily placed in the simulation box with a separation over 5 Å. These GLRX monomers were seen forming some metastable asymmetrical interfaces in our simulations.

Comment 16: The MD simulations only demonstrate the importance of C8-C8 and C23-C8 interactions for the formation of the dimer, what about C83? And C79 (see the Hashemy et al. paper mentioned above)

Response 16: Since C79 is buried, we did not see the C79 residues from two GLRX closer than 12 Å in any frame of our simulations. We also did not see the distance of C79 to other cysteines closer than 12 Å. C83 is on the protein surface. However, we did not see a stable interface formed between C83 and cysteine from another GLRX. Additionally, Hashemy et al observed C79 and 83 dimer formation upon oxidation of GLRX by incubation with GSSG. Our simulations observed dimer formation with reduced GLRX.

Comment 17: It is hard to understand {quality of the print?} only two monomers/two different colors are visible in the fig. Maybe also include a zoomed out figure of the structure as well? To get a general idea about the structure of the dimers?

Response 17: We have adjusted the Figure to improve the image. The new Figure 6A has shown both the symmetrical and asymmetrical interfaces.

Comment 18: Lines 417-420 On what is the conclusion of the "slightly different dynamics" based, especially since the comparison of the abundance is not possible?

Response 18: We thank the reviewer for their comment and agree that the language used was too strong. We have clarified the language describing the results of the mass spectrometry experiments and now discuss area under the curve measurements which permits abundance comparisons among the same peptide for ascending GSSG concentrations. The body of the text has been revised as follows: "Although comparisons of abundance between different tryptic peptides are not feasible due to differences in ionization efficiencies of each peptide, these results confirm that glutathionylation of each cysteine in WT GLRX occurs following incubation with GSSG, and suggest that C23, C26, and C8 are more readily glutathionylated at lower concentrations of GSSG".

Comment 19: [Fig 5B] Only C8 and C83 mutants mentioned as those that remained their activity after the incubation with GSSG. what about C26? the reduction of the activity was either not significant or the significance was not included in the figure.

Response 19: We apologize for this oversight. We highlighted the role of C8 and C83 mutations in protecting against GSSG- induced inactivation. We did conduct statistics for all mutants. The activity of Cys26S GLRX was not statistically significantly decreased in response to GSSG. We have highlighted this in the revised Figure 4C via the insertion of NS (not significant). We have edited the manuscript text to include effect of mutation of C26 in the result section.

Comment 20: [Fig 7C] Why do the triple mutants with C8 (C8/26/79S and C8/26/83S) show no reduction of oligomerisation as the single and quad mutant?

Response 20: We are not certain why the triple mutants show a different oligomerization pattern compared to the single C8 or quad mutant. It is worth noting that all mutants still contain the active site Cysteine 23. Molecular modeling showed that C23 can be involved in dimer formation (New Figure 6A). Besides oxidation of cysteines, H₂O₂ can also oxidize other amino acid residues, including methionines, which we did observe in our Mass Spectrometry studies (data not shown). We speculate that H₂O₂ induced-oxidation of other amino acids, or of C23, is contributing to the observed oligomerization or dimerization, in the case of the quad mutant. However, addressing these putative scenarios will need additional structural characterization of each of these mutants.

Comment 21: [Fig 7C] It seems like the ~20kDa band consist out of two bands. The C8/26/79S mutant shows clearly two bands: 20 and ~22 kDa. Where does the second band come from? (see also Fig. 5)

Response 21: We agree with the reviewer that this banding pattern is quite striking. As mentioned above, these samples were prepared and run under aerobic conditions. We speculate that the banding pattern is the result of different formation of intra- or inter-molecular disulfides. Following incubation of GLRX with GSSG (new Figure 6D) additional bands observed are likely the result of a mass shift from the addition of GSH to GLRX.

Reviewer #2

Comment 1: The manuscript of Elizabeth Corteselli et al. is unfortunately very disappointing and additional experiments are needed before it potentially could be accepted for publication. Further the way that the study it is currently written is rather basic, as the authors present a collection of observations and then the discussion does not really go deeper.

One major point is the physiological significance of all the experiments. Authors are really using huge GSSG concentrations that will never occur in the cytosol - can you really speak about "hierarchies" then?

Response 1: We sincerely regret that the reviewer was not enthusiastic about the writing style and the data presented in the manuscript. To address his/her/their concern, we have substantially rewritten the manuscript to ensure improved integration between the in silico and experimental studies. We do believe that this manuscript addresses important new insights into the functioning of GLRX, namely its inactivation by its own substrate, GSH, using an assay that was modified for this purpose. We respectfully believe strongly that this is a critical area of investigation and that our data provide an important advance in knowledge based on the essential premise that GLRX oxidative inactivation happens in a number of disease scenarios. Therefore, improved understanding at the molecular level about the oxidative vulnerabilities of each of GLRX's cysteines (explored via sophisticated modeling and mutational validation) is highly relevant. As mentioned in our response to reviewer 1, we do not believe that the concentrations of GSSG used in this study would occur in the cytosol. Induction of glutathionylation in the test tube is well known to require large concentrations of GSSG, in agreement with several published manuscripts by others in the field, and that GSSG is the most direct chemical way to induce glutathionylation (PMID: 17355958, PMID: 34399606, PMID: 34743784, PMID: 15657297).

Comment 2: Can glutathionylation really be protective as they suggest as GSH concentrations remain high after "oxidative insults" (see previous studies)?

Response 2: The reviewer is correct that GSH is present in high concentrations in the cytosol. However, glutathionylation does indeed occur intracellularly despite high concentrations of GSH (PMID: 33817836, PMID: 32971362, PMID: 29988126, PMID: 27035878, PMID: 16515838, PMID: 34876574, PMID: 16916935). Redox-based signaling is proposed to occur through redox scaffolds or localized actions of NOX or GST enzymes to potentially create microenvironments with locally higher concentrations of oxidants that favor glutathionylation (PMID: 20690882, 26067716). However, we do not claim that glutathionylation is a universally protective feature. While it has been shown that glutathionylation can chemically protect a protein from overoxidation (ie sulfenic acid to sulfinic and sulfonic acid) (PMID: 31813265), there are also documented cases where glutathionylation results in biologically detrimental effects that are target protein-dependent, such as loss of enzymatic activity as shown in this manuscript, or gain of biological functions of other proteins (PMID: 19171757).

Lastly, we again wish to highlight again that the scientific premise for our in-depth investigation of the cysteines targeted for oxidative inactivation of GLRX stem from our observations in lung tissues for patients with idiopathic pulmonary fibrosis wherein GLRX activity was substantially lower, in association with its S-glutathionylation (PMID: 29988126). We therefore believe that there is considerable importance of efforts to unravel the importance of oxidative inactivation of GLRX and the cysteines involved. We strongly believe that our manuscript delivers in this regard through the incorporation of an assay that measures substrate de-glutathionylation by GLRX (the first reaction step) in silico modeling of the reactivities of each of the cysteines of GLRX towards H₂O₂ or GSH, and experimental validation using Cys-Ser mutations of each of the cysteines of GLRX alone or in combination.

Comment 3: No mention of GR and kinetic competition with it. Basically, a lot of critical points...

Response 3: We did not mention glutathione reductase in this study, as the focus of our manuscript was on the activity and oxidative inactivation of glutaredoxin. Therefore, we did not include GR in the assay. While GR catalyzes the reduction of GSSG, we do not propose that GR and GLRX would be in competition for the diE-GSSG substrate. In fact, it was previously shown by Coppo et al that GR does not act on diE-GSSG, while it is an excellent substrate for GLRX (PMID: 26836485). Further, we also conducted studies with glutathionylated BSA (to model a physiological substrate for GLRX), which would not be a potential substrate for GR. Reviewer 1 also raised concerns around the assay conditions which is why we have inserted new Figure 1D in our revised manuscript in order to clarify the assay conditions.

Suggestion: Authors might have a look at the review of Marcel Deponte 2017.

Response: We thank the reviewer for mentioning this important manuscript, of which we are well aware. We have now included it among our references and regret to omit this in the earlier version.

Comment 4: How to improve the manuscript: A thorough comparative kinetic stopped flow analysis with GR could be a good starting point.

Response 4: We thank the reviewer for this important suggestion and agree that this would provide valuable data. This will be an area of further focus to comparatively assess WT GLRX and the 11 mutant versions that we describe herein. At this time, we did not have the methodology and bandwidth to conduct these experiments but hope to expand upon our current observations with collaborating scientists in the future. Further, as stated in response 3, GR is not present in the current manuscript as the original assay developed by Coppo and colleagues (PMID: 26836485) was modified herein precisely to overcome potential issues with competition between oxidatively modified GLRX and NADPH/GSH/GR.

Reviewer #2 (Remarks to the Author):

Although the authors have made a commendable effort to address many of the comments, I remain apprehensive about the genuine novelty of this study. However, from a scientific standpoint, the story appears to be sound. While I have no additional comments, I did notice that the discussion on nucleophilicity overlooked the Brønsted theory. It is important to note that a decrease in the pKa of a Cys residue does not necessarily correlate with an increase in its nucleophilicity. This is particularly relevant when the pKa drops far below the operational pH range.

Reviewer #3 (Remarks to the Author):

GLRX are important oxidoreductases that can regulate PSSG. Elizabeth M. Corteselli and her colleagues have integrated computational and experimental approaches to investigate the functions of five cysteines in GLRX and found that C8 and C83 are key targets for S-glutathionylation. In general, I feel that the calculation part is generally reasonable, but the presentation of the results is very rough and some details still need to be supplemented before it potentially could be accepted for publication. Hope these suggestions can help the authors refine their article.

1. The initial position and distance between the two proteins in the dimer have great influences on the simulation results, especially in the relatively small simulation scale (200-400 ns in this article). The authors should describe the construction process in detail, rather than citing previous literature.
2. Did the authors carry out a standard pretreatment for the assemblies before producing simulation results? For example, we need to carry out energy minimization for the assembly first; then restrict the protein and equilibrate the water molecules for 100-500 ps; and finally, equilibrate the whole assembly without restraint for 1-5 ns. Please refer to the literature of Gregg T. Beckham (J. Am. Chem. Soc. 2014, 136, 8810–8819).
3. 310 K is more suitable than 300 K, if we plan to use this method as a treatment. Because 310 K is closer to the in vivo environment.
4. According to the current technical conditions, I personally think that the simulation scale of monomers (19,000-28,000 atoms) should reach the microsecond level.
5. Why is the difference in the number of atoms between monomers so large? For example, the C83S mutation leads to more than 8000 atoms difference, compared with the wild type. Does the size of the solvent boxes guarantee a minimum distance (such as 15 Å) from any atom of the protein to any edge of the simulation box?
6. The authors should describe the QM area in more detail. For instance, the total charge and the total atom numbers. Furthermore, the authors should also provide the movie for the reaction in the QM area.
7. In Figure 3, the stick models should be refined. (change double bond into single bond; change the hydrogen scale to 1.0)
8. The residues in Figure 5 are shown in ball-and-stick models, but in Figure 3 and 6 they are stick models. The full text should be unified.
9. In the videos, it is better to display the protein by the cartoon model instead of the existing cartoon and stick models.
10. In video 2, the orientation of the protein should be fixed, not rotating.

Point-by-point responses to the reviewers' comments:

Reviewer #2:

Comment: Although the authors have made a commendable effort to address many of the comments, I remain apprehensive about the genuine novelty of this study. However, from a scientific standpoint, the story appears to be sound. While I have no additional comments, I did notice that the discussion on nucleophilicity overlooked the Brønsted theory. It is important to note that a decrease in the pKa of a Cys residue does not necessarily correlate with an increase in its nucleophilicity. This is particularly relevant when the pKa drops far below the operational pH range.

Reply: We regret that reviewer 2 remains somewhat apprehensive about the novelty of our findings. We delineated in our previous rebuttal and in the revised manuscript that despite prior knowledge of oxidative inactivation of GLRX, the exact contributions of each GLRX cysteines to oxidative inactivation remained unknown and formed the scientific premise for the present study. We reiterate that a major contribution of our manuscript is the elucidation of the role that each individual cysteine plays in inactivation as measured by a more specific and physiologically relevant enzyme assay system, advances that are relevant to the broad research community. We applied for the first time a combined strategy consisting of molecular simulations and hybrid QM/MM calculations to gain molecular understanding of the structure-function relationships of all GLRX cysteine residues. This combinatorial information, along with experimental validation, in our opinion presents a substantial advancement of knowledge in our understanding of the oxidative vulnerabilities of this critical redox enzyme.

We thank Reviewer #2 for the comment regarding nucleophilicity. In our estimate, the pK_a of the five cysteine residues in GLRX falls in a wide range from 5 to 17. While we agree that the pK_a of a cysteine residue does not necessarily correlate with an increase in its nucleophilicity, C23 is the only cysteine with $pK_a < 7$ in GLRX. Our finding of C23 as the most acidic among all five cysteine agrees with previous reports that it plays a key role in GLRX's enzymatic activity [47]. This is also consistent with prior observation that the catalytic cysteine residues are acidic in class 1 glutaredoxins (PMID: 30593501).

Reviewer #3:

GLRX are important oxidoreductases that can regulate PSSG. Elizabeth M. Corteselli and her colleagues have integrated computational and experimental approaches to investigate the functions of five cysteines in GLRX and found that C8 and C83 are key targets for S-glutathionylation. In general, I feel that the calculation part is generally reasonable, but the presentation of the results is very rough and some details still need to be supplemented before it potentially could be accepted for publication. Hope these suggestions can help the authors refine their article.

Comment 1: The initial position and distance between the two proteins in the dimer have great influences on the simulation results, especially in the relatively small simulation scale (200-400 ns in this article). The authors should describe the construction process in detail, rather than citing previous literature.

Reply 1: We regret that we did not sufficiently explain the procedures and that the presentation of the data was perceived as rough. We have made changes in the text to address these. We have clarified in the text how we built the initial models of multiple GLRX proteins as follows:

Page 33, Line 784-790: The MD simulation of GLRX dimer (monomeric conformation PDBID: 4RQR) aligned to zebrafish GLRX (PDBID: 3UIW) represents the symmetrical C8-C8 interface (shown in surface mode on Supplementary Figure 8) which is driven mainly by polar interactions. In addition, three GLRX monomers (PDBID: 4RQR) were arbitrarily placed in the simulation box with a separation over 5 Å. All the models were built in Maestro (Schrödinger, Inc.) and prepared for simulation in System Builder.

Comment 2: Did the authors carry out a standard pretreatment for the assemblies before producing simulation results? For example, we need to carry out energy minimization for the assembly first; then restrict the protein and equilibrate the water molecules for 100-500 ps; and finally, equilibrate the whole assembly without restraint for 1-5 ns. Please refer to the literature of Gregg T. Beckham (J. Am. Chem. Soc. 2014, 136, 8810–8819).

Reply 2: We thank the reviewer for this comment. We have used the multistage equilibrium strategy in the Desmond package, as follows. This standard strategy has been shown effective in many prior publications including ours.

stage 1 - task
stage 2 - simulate, Brownian Dynamics NVT, T = 10 K, small timesteps, and restraints on solute heavy atoms, 100 ps
stage 3 - simulate, NVT, T = 10 K, small timesteps, and restraints on solute heavy atoms, 12 ps
stage 4 - simulate, NPT, T = 10 K, and restraints on solute heavy atoms, 12 ps
stage 5 - solvate_pocket
stage 6 - simulate, NPT and restraints on solute heavy atoms, 12 ps
stage 7 - simulate, NPT and no restraints, 24 ps
stage 8 - simulate
(8 stages in total)

To improve the clarity, we added a sentence to **Page 34, Line 790:** The multistage equilibrium strategy in the Desmond package was used.

Comment 3. 310 K is more suitable than 300 K, if we plan to use this method as a treatment. Because 310 K is closer to the *in vivo* environment.

Reply 3: We thank the reviewer for pointing this out. We agree that 310 K is closer to the *in vivo* environment. However, we did not see a significant temperature effect between 300 and 310 K for GLRX simulations. To be consistent with the *in vitro* experiments in this work, we decided to report our simulation data at 300 K.

Comment 4. According to the current technical conditions, I personally think that the simulation scale of monomers (19,000-28,000 atoms) should reach the microsecond level.

Reply 4: We agree with Reviewer #3 that it is technically feasible to extend the monomer simulations to one microsecond. However, our analysis of our monomer simulations (400 ns in length) shows small RMSD fluctuations (Supplementary Figure 5, see below), convergence in the energy (especially in the electrostatic energy), as well as little change in surface area and radius of gyration. These data seem to suggest the sampling is sufficient within the 400 ns simulation time.

Supplementary Figure 5. Time evolution of protein Ca RMSD in our MD simulations. These data show that mutation of cysteines to serine does not change the tertiary structure of GLRX. RMSD of 400 ns MD simulation of i. GLRX wildtype with C23-C26 disulfide bond (black circle) and reduced WT GLRX (orange square). ii. GLRX C8S mutant, iii. GLRX C83S mutant, and iv. GLRX C79S mutant.

Comment 5. Why is the difference in the number of atoms between monomers so large? For example, the C83S mutation leads to more than 8000 atoms difference, compared with the wild type. Does the size of the solvent boxes guarantee a minimum distance (such as 15 Å) from any atom of the protein to any edge of the simulation box?

Reply 5: In response to your concern, we have re-examined our protocol of model construction using System Builder (Schrödinger, Inc.). The atom difference mainly comes from the size of the box, as the smaller constructs were in the orthorhombic box. Most of the simulations were performed in the cubic box. There should be no difference in our simulations, with a buffer distance of 10 Å to ensure a sufficient simulation box.

Comment 6. The authors should describe the QM area in more detail. For instance, the total charge and the total atom numbers. Furthermore, the authors should also provide the movie for the reaction in the QM area.

Reply 6: We thank Reviewer #3 for this comment. We endeavor to provide full details of our QM/MM calculations to readers. The QM coordinates are included on **Page 36 to 50**. In our QM/MM calculations, the QM region has a total charge of -1 and the total number of atoms varies from 33 to 49. This difference is due to the sequence difference, as we include the residue before and after the target cysteine in the QM region.

While generating a movie of the reaction may require ab initio MD or other visualization techniques, we would like to provide a free energy profile to illustrate the model reaction (Supplementary Figure 4, see below). Cysteine residues in GLRX display much lower reaction barriers, as shown in Figure 5.

Supplementary Figure 4. Free energy profile of the reduction of H₂O₂ by model compound [CH₃S...HOOH]– using B3LYP/6-31G* level of theory.

Comment 7: In Figure 3, the stick models should be refined. (change double bond into single bond; change the hydrogen scale to 1.0)

Reply 7: In response to this concern, we have updated Figure 3 (see below) with rescaled hydrogen stick representation, and the double bond representation was kept for clarity.

Figure 3. Molecular docking reveals hierarchy of reactivity of GLRX cysteines with GSSG. The docking pose corresponds to the lowest S-S distance of GSSG docked to the proximity of **A.** C8. **B.** C23. **C.** C26. **D.** C79 and **E.** C83. The sulfur groups on cysteines are shown as yellow spheres and the GSSG ligand is shown in sticks. The yellow dashed lines show the S-S distances

labelled with numbers in Å. Inset: *The reported scores are the ones of the poses representing the lowest distance between S-S groups. **All numbers in Panels A - E are distances in angstrom.

Comment 8: The residues in Figure 5 are shown in ball-and-stick models, but in Figure 3 and 6 they are stick models. The full text should be unified.

Reply 8: We apologize to the reviewer for the inconsistency. We have updated Figure 5 (see below) to be consistent in the style with Figure 3. The H₂O₂ model is kept as balls and sticks, to be distinguished from the stick representation of the protein.

Figure 5. Quantum Mechanics/Molecular Mechanics (QM/MM) modeling of GLRX cysteines reveal that C8 and C83 are most reactive with H₂O₂. Transition state of GLRX/H₂O₂ system for all the cysteines and representation of the stabilizing interactions in protein environment. **A.** C8. **B.** C23. **C.** C26. **D.** C79 and **E.** C83. The labels of dash lines show distances in Å. Inset: E_a and

ΔH for each cysteine residue are indicated. **F.** Mutation of GLRX cysteine 8 and 83 protects against oxidative inactivation from H_2O_2 . GLRX mutants were incubated with $100 \mu M H_2O_2$ for 10 minutes and H_2O_2 was removed by treatment with catalase prior to measurement of activity with diE-GSSG. Results are expressed as percentage of reduced activity compared to untreated controls for each mutant. Mean \pm SD, $n = 3$. Statistics performed using multiple t-tests between % remaining activity and reduced activity (100%) for each mutant, * $p \leq 0.05$, ** $p \leq 0.01$, *** $p \leq 0.001$, **** $p \leq 0.0001$.

Comment 9: In the videos, it is better to display the protein by the cartoon model instead of the existing cartoon and stick models.

Comment 10: In video 2, the orientation of the protein should be fixed, not rotating.

Reply 9 and 10: We have remade video 1 and video 2, with the updated illustration.

Video 1: This video illustrates the stability of the symmetric C8-C8' interface. This 200-ns MD simulation starts from two GLRX monomers (PDB ID: 4RQR, gold and cyan cartoons) aligned to a zebrafish Grx2 dimeric structure in solution. The C8 sidechains are shown in sphere and labelled.

Video 2. This video illustrates the formation of the asymmetric C8-C23' interface. This 200-ns MD simulation starts with free GLRX monomers (PDB ID: 4RQR, gold and skype blue cartoons) in solution. The C8 and C23 sidechains are shown in sphere and labelled.

Reviewer #3 (Remarks to the Author):

The authors have revised their article following most of the suggestions. I have no additional comments.